# Bridging the gap: A new species of arboreal *Abronia* (Squamata: Anguidae) from the Northern Highlands of Chiapas, Mexico

**Adam G. Clause**[1]*, **Roberto Luna-Reyes**[2⊙], **Oscar M. Mendoza-Velázquez**[3⊙], **Adrián Nieto-Montes de Oca**[4⊙], **Israel Solano-Zavaleta**[5⊙]

1 Department of Herpetology, San Diego Natural History Museum, San Diego, California, United States of America, 2 Dirección de Áreas Naturales y Vida Silvestre, Secretaría de Medio Ambiente e Historia Natural, Tuxtla Gutiérrez, Chiapas, México, 3 Instituto de Ciencias Biológicas, Universidad de Ciencias y Artes de Chiapas, Tuxtla Gutiérrez, Chiapas, México, 4 Departamento de Biología Evolutiva, Facultad de Ciencias, Universidad Nacional Autónoma de México, Ciudad de México, México, 5 Departamento de Ecología y Recursos Naturales, Facultad de Ciencias, Universidad Nacional Autónoma de México, Ciudad de México, México

⊙ These authors contributed equally to this work.
* adamclause@gmail.com

**Data Availability Statement:** All relevant data are within the manuscript, except for the exact GPS coordinates where specimens were collected;

## Abstract

The mountain forests of Middle America are renowned for their endemic biodiversity, and arboreal alligator lizards (genus *Abronia*) are high-profile vertebrates endemic to this region. In this work, we describe a new species of arboreal *Abronia* that is known only from the type locality in the Northern Highlands of Chiapas, Mexico. The new species is diagnosed from all other members of the genus *Abronia* by the following combination of characters: lack of protuberant or spine-like supra-auricular scales, lack of protuberant or casque-like postero-lateral head scales, dorsum of head pale yellow with distinct dark markings, 35–39 transverse dorsal scale rows, lateralmost row of ventral scales enlarged relative to adjacent medial row, and dorsum brown with darker crossbands that are sometimes reduced to rows of spots. We provisionally include the new species in the subgenus *Lissabronia* based on genomic and morphological evidence, but our results also suggest a close relationship to the subgenus *Abaculabronia*. The new species is geographically separated from the nearest *Lissabronia* and *Abaculabronia* species by the lowland Central Depression of Chiapas. Ongoing habitat loss and other factors imperil the new species, leading us to propose its listing under multiple threatened species frameworks. Because the Northern Highlands have poor coverage of protected areas, we briefly comment on the potential of this new species for stimulating conservation in the region.

## Introduction

The highland forests of Middle America are biodiversity storehouses roughly analogous to islands [1, 2]. These cool, wet woodlands form an archipelago of separate patches surrounded by seas of lowland vegetation [3, 4]. This isolation has fostered the evolution of remarkable organismal diversity [5]. Hypothesized causes of this diversity remain poorly tested, but

those data are held by the public museums where the specimens are deposited, and are not released in the manuscript due to conservation concerns that we detail therein. Contact information for the institutional body to which data request for the withheld information may be sent is as follows: Dr. Uri Omar García-Vázquez urigarcia@gmail.com Professor and Curator Museo de Zoología, Facultad de Estudios Superiores Zaragoza Universidad Nacional Autónoma de México.

**Funding:** Collection of molecular data for this study was funded by a grant from the Dirección General de Asuntos del Personal Académico, Universidad Nacional Autónoma de México (PAPIIT no. IN218522) to ANMO.

**Competing interests:** The authors have declared that no competing interests exist.

interrelated biological, geological, and climatic factors are likely at play [5, 6]. Many authors generally agree that complex historical processes of mountain formation, together with paleo-climatic events, created biogeographic barriers correlated with speciation in many Middle American vertebrates [7–13]. Today, striking and emblematic species of birds [14, 15], snakes [16, 17], frogs [18, 19] and salamanders [20, 21] live in these mountain woodlands and nowhere else. Many reptile and amphibian groups adapted to these forests have notably rapid rates of species turnover across short distances, with species often being known only from a single isolated peak or massif [18, 22–27]. These high rates of endemism, coupled with pressing threats such as climate change and rapid deforestation caused by humans [28–31], have generated much conservation interest in the highlands of Middle America [32–35].

One conservation-relevant group that has diversified across these "sky island" forests is the alligator lizard genus *Abronia* Gray, 1838 [36] (Anguidae: Gerrhonotinae). Although a record exists from California, United States of an extinct Miocene species referred to this genus [37], all known living *Abronia* cumulatively range from central and eastern Mexico southward to western Panama [38]. These lizards are almost always restricted to humid highland forests, with evergreen cloud forests and seasonally dry pine-oak forests being particularly common habitats [39]. Traditionally the genus included only arboreal species, although arboreality is a presumption for a few species because no ecological data exists for them [39–42]. Recent genomic analyses, however, indicate that this assemblage of arboreal and probably arboreal species is a paraphyletic group with respect to the strictly terrestrial genus *Mesaspis* Cope, 1877 [38, 43]. Because *Abronia* has nomenclatural priority over *Mesaspis*, the authors of that study recommended the synonymization of the latter [38]. We follow that taxonomic arrangement in the present study. More comprehensive taxonomic histories of *Abronia* and the former genus *Mesaspis* are available elsewhere [38, 44, 45]. Presently, there are 41 recognized species of *Abronia* [45]. These are divided into 11 clades or species groups. Eight of these are composed exclusively of arboreal species, while the remaining three include only terrestrial species [38, 39]. The arboreal species (which constitute a three-fourths majority of the genus) are often colorful, iconic, and imperiled [46]. Most arboreal *Abronia* species are also mysterious, with limited geographic distributions and cryptic behavior that combine to produce few sightings [39]. These factors have made arboreal *Abronia* a high-profile target for international biodiversity management [47–49]. Discovery of previously unknown species of the genus *Abronia* is thus likely to motivate strong conservation interest.

In Mexico, the mountains of the southernmost state of Chiapas remain especially promising for undocumented herpetofaunal diversity. These areas of Chiapas recently yielded range extensions and the first records for Mexico of a snake and a rarely-seen salamander [50, 51], plus an endemic arboreal *Abronia* species entirely new to science [44]. A likely region for additional such finds is the underexplored system of massifs in the north-central part of the state. These complex mountains are generally identified as part of the Northern Highlands (Montañas del Norte) physiographic region [52–56]. However, they have also been called the Central Highlands [57], or considered part of the Meseta Central [18], or part of Los Altos de Chiapas [32]. Regardless of what these highlands are called, only two arboreal *Abronia* are reported to exist there. One is an unallocated population of the *Scopaeabronia* clade from the Zona Sujeta a Conservación Ecológica "Laguna Bélgica," in the southwestern part of the region [38, 39, 58, 59]. The second is *A. lythrochila*, which belongs to the *Auriculabronia* clade and barely enters the Northern Highlands along a southern escarpment near the town of Jitotol [60–62]. Most of the Northern Highlands is thus a gap on the arboreal *Abronia* distribution map.

In 2014, intriguing photographs emerged of an arboreal *Abronia* from this gap. We made subsequent collecting expeditions in 2015, 2021, and 2022, and eventually gathered sufficient

comparative material. Based on an integrative analysis of genomic, morphological, and bio-geographical evidence, in this study we delimit that population as a new species. Remarkably, this new species belongs to neither the *Scopaeabronia* nor *Auriculabronia* clades. Instead, we find that it is closely related to *A. morenica*, which occurs some 110 km away on the opposite side of the inhospitable lowland Central Depression of Chiapas [44]. We also comment on the biogeographical and conservation implications of the new species.

## Materials and methods

### Field specimen collection

During our multi-year fieldwork in the Northern Highlands of Chiapas, Mexico, we captured five specimens of the presumed new species of arboreal *Abronia*. All specimens originated from near the town of Coapilla. We searched for specimens on foot and by climbing trees, and we captured them by hand or by a lasso attached to a telescoping pole. We euthanized speci-mens with an intracardiac injection of sodium pentobarbital within five days of field capture, gathered a liver sample for later molecular analyses (see below), then fixed them in a 10% dilu-tion of full-strength buffered formalin, followed by transfer to 70% ethanol for permanent stor-age. We deposited the specimens in the herpetological collection of either the Museo de Zoología "Alfonso L. Herrera" of the Facultad de Ciencias, Universidad Nacional Autónoma de México (MZFC-HE, formerly MZFC) or the Museo de Zoología of the Facultad de Estudios Superiores Zaragoza, Universidad Nacional Autónoma de México (MZFZ). Additionally, we examined relevant specimens housed in the Museo de Zoología of the Universidad de Ciencias y Artes del Estado de Chiapas (MZ-UNICACH) to confirm the reptile and amphibian assem-blage that coexists with the new species. Institutional Animal Care and Use Committees do not exist in Mexico nor at the institution of the lead author, but all of our live animal handling procedures followed the recommendations available in the Guidelines for Use of Live Amphib-ians and Reptiles in Field and Laboratory Research [63]. Specimen collection was authorized under permit FAUT-0093 (SGPA/DGVS/4755/19) issued by the Secretaría de Medio Ambiente y Recursos Naturales to Adrián Nieto-Montes de Oca, and permit FAUT-0243 (SGPA/DGVS/03937/21) issued to Uri Omar García-Vázquez. The sites that we visited during fieldwork are communally owned by the Ejido de Coapilla, and we are grateful to the Comisari-ado Ejidal (David Cruz-Pérez) and other residents of Coapilla for authorizing our access to their lands for this study.

### Molecular procedures

**ddRADseq libraries.** To investigate the distinctness and phylogenetic relationships of the presumed new species using molecular evidence, we gathered ddRADseq data previously pub-lished for representative taxa of the main clades of *Abronia* [38, 45] and expanded this dataset with new ddRADseq data from samples of three taxa previously absent from molecular phylo-genetic hypotheses of the genus: two specimens of the presumed new species from Coapilla, the holotype and two paratype specimens of *A. morenica*, and one paratype of *A. ornelasi*. We also generated new ddRADseq data for two specimens of *Gerrhonotus* and one of *Elgaria*, as well as one of *A. antauges* (S2 Appendix). Our sampling included all *Abronia* taxa that occur within 200 km of the presumed new species.

We generated the new ddRADseq data following a recently published procedure used for *Abronia* [45], except that we extracted genomic DNA from the presumed new species, *Gerrho-notus*, and *Elgaria* samples with a DNeasy Blood & Tissue Kit (Quiagen, cat. No. 69504). Also, we sequenced the samples of the presumed new species using a 150 bp paired-end HiSeqX platform (Macrogen Inc., Korea), whereas we sequenced the *Gerrhonotus* and *Elgaria* samples

using a 100 bp paired-end HiSeq platform at Arizona State University. All novel sequence data are available on GenBank (BioProject accession number PRJNA 1034133; S2 Appendix).

We processed the ddRADseq dataset using ipyrad ver. 0.9.50 [64]. We used default parameter settings to complete assemblies, except for the following settings: maximum number of barcode mismatches = 1; filter for adapters/barcodes = 2 (strict); both minimum depth at which statistical and majority rule base calls are made during consensus base calling = 10; and clustering threshold = 0.94 [38, 45]. In addition to using the default setting of a minimum number of samples that must have data at a given locus for it to be retained in the final assembly (= taxon coverage) of 4, we generated an assembly with taxon coverage = 17 (approximately one-half of the total number of samples, n = 35) to explore the effect on robustness of phylogenetic analyses of different numbers of loci and different percentages of missing data.

**Phylogenetic, population genetic, and species delimitation analyses.** We performed maximum likelihood (ML) analyses using RAxML ver. 8.2.12 [65] of the two matrices with different taxon coverage (see above). The matrices included all concatenated loci with SNPs and invariant sites to improve branch length and topological accuracy in phylogenetic reconstructions [66]. We performed a simultaneous search to obtain the best-scoring ML tree and a rapid bootstrap analysis with the GTR + GAMMA model, using 100 bootstrap replicates starting from random addition sequence trees. We performed all of the analyses on the Mana high performing computing cluster of the University of Hawaii. We used FigTree ver. 1.4.4 [67] to produce the figure with the resulting phylogenetic tree.

We characterized population genetic structure within the lineages in the clade composed of the presumed new species, *A. morenica*, and *A. ornelasi* (see below) using conStruct ver. 1.0.5 [68]. The conStruct software is a relatively new statistical method for the simultaneous inference of continuous and discrete patterns of population structure. We chose this method because isolation by distance, a pattern that is continuously distributed across a landscape, is widespread in nature and because models of discrete population structure may incorrectly (especially when sampling is discontinuous) ascribe differentiation due to continuous processes such as isolation by distance to discrete processes such as geographic, ecological, or reproductive barriers between populations [69]. Because conStruct is sensitive to missing data, we first generated an assembly of SNPs (one per locus) in ipyrad ver. 0.9.50 [64] with only those loci with data from all the samples (6) and other parameter settings as outlined above. We then performed a cross-validation analysis with two repetitions, number of layers (K) = 1–4, two chains, and 30000 iterations to determine the statistical support for models with different numbers of layers with and without a spatial component, and finally performed an analysis for the model with the highest statistical support with two chains and 30000 iterations.

We performed a species delimitation analysis on the clade composed of the presumed new species, *A. morenica*, and *A. ornelasi* (see below) using the heuristic criterion for species delimitation based on a genealogical divergence index (gdi) between populations [70]. First, we generated an assembly in ipyrad ver. 0.9.50 [64] with only those loci for which all the samples in the assembly (6) had data, and other settings as outlined above. Then we performed a A00 analysis in the program BPP ver. 4.6.2 [71, 72] to estimate the parameters of species divergence times and population sizes under the multispecies coalescent model, and finally used the posterior means of the parameters generated by BPP to calculate the gdis [70]. Based on the mean uncorrected pairwise genetic distance (p-distance) within and among the taxa, we assigned the population size parameters (Øs) the inverse-gamma prior IG(3, 0.0016), with mean 0.0016/(3–1) = 0.0008; based on the mean p-distance between the basal-most taxon (*A. ornelasi*) and the other taxa, we assigned the divergence time at the root of the species tree ($TAU_0$) the inverse-gamma prior IG(3, 0.0016), with mean 0.008; we used the uniform Dirichlet distribution to specify the other divergence time parameters [73]. We performed the A00 analysis with data

from 1000 loci. We ran the analysis for 1,000,000 generations with burn-in of 40,000 generations and sampling every 200 generations. To verify convergence, we used Tracer ver. 1.7.2 [74]. We ran the analysis twice to confirm consistency between runs.

## Morphology

We consulted standard reference works for scale terminology [75] and for scale count protocols [39, 76]. We scored bilateral characters on both sides and give the conditions on the left and right sides as left/right. For counts of transverse dorsal and transverse ventral scale rows, we express these counts as a range for specimens with aberrant fission/fusion of scale rows. After euthanasia, but prior to formalin fixation, we took linear measurements with dial calipers (to the nearest 1 mm), and we measured mass with a digital pan scale (nearest 0.1 g).

We based our diagnosis and comparisons with other members of the genus *Abronia* on a review of the relevant descriptive literature, with details available elsewhere [45]. To promote comparison with earlier literature, the format of our description largely follows those in the latest monograph on *Abronia* [39]. Of the 11 clades or species groups of *Abronia* recognized by the most recent morphological treatment [39] and/or molecular treatment [38], all are supported by morphological characters and all except three are also supported by molecular data. However, four of these groups lack a formal name, and the terrestrial clades are all morphologically similar. For ease of comparison in the diagnosis, we thus refer to the six named species groups (subgenera) of arboreal *Abronia* that have been traditionally recognized [39], and we refer to all terrestrial *Abronia* as "former members of the genus *Mesaspis*." Although this arrangement imperfectly reflects the evolutionary history within *Abronia* [38], these groups are morphologically cohesive and easily diagnosable [39]. Hence, this arrangement facilitates the simplest, most straightforward comparison of the presumed new species with all existing congeneric species. Additionally, in this study we adopt the evolutionary species concept [77–79] and follow an integrative approach that uses genetic distinctiveness, fixed differences in morphological features [80] and geographical isolation to delimit the existence of distinct species-level lineages.

## Nomenclatural act

The electronic edition of this article conforms to the requirements of the amended International Code of Zoological Nomenclature, and hence the new names contained herein are available under that Code from the electronic edition of this article. This published work and the nomenclatural act it contains have been registered in ZooBank, the online registration system for the ICZN. The ZooBank LSID (Life Science Identifier) can be resolved and the associated information viewed through any standard web browser by appending the LSID to the prefix http://zoobank.org/. The LSID for this publication is urn:lsid:zoobank.org:pub:5E56D78F-1DEF-42B3-AEBD-B2AD2E1D1817. The electronic edition of this work was published in a journal with an ISSN, and has been archived and is available from the LOCKSS digital repository.

## Results

The total numbers of raw reads that we obtained from the ddRADseq libraries generated for this study (see above) were as follows: 4195570 and 3557445 for the two samples of the presumed new species from Coapilla (MZFC-HE 36544 and 36545, respectively); 2388811, 3053286, and 2317826 for the three samples of *A. morenica* (MZFC-HE 33486, 33487, and 33490, respectively); 130062 and 3552001 for the samples of *A. antauges* (MZFC-HE 29310) and *A. ornelasi* (UTA R-12499), respectively; and 1876800, 2771566, and 3267734 for the

samples of *Gerrhonotus liocephalus* (MZFC-HE 16988), *G. mccoyi* (MZFC-HE 29648), and *Elgaria kingii* (ANMO 4163), respectively.

The assemblies composed of loci with data from at least 4 and 17 samples had sequences from 27527 and 3425 loci, 83388 and 22000 parsimony-informative sites, sequence matrix sizes of 3846667 and 488700 sites, and 74.18% and 39.18% of missing sites, respectively.

The phylogenetic hypotheses recovered from the ML analyses of the two datasets were nearly identical (Fig 1) and all their relationships were statistically supported (i.e., support values ≥ 75) except for those among the *A. morenica* samples. The hypotheses recovered the

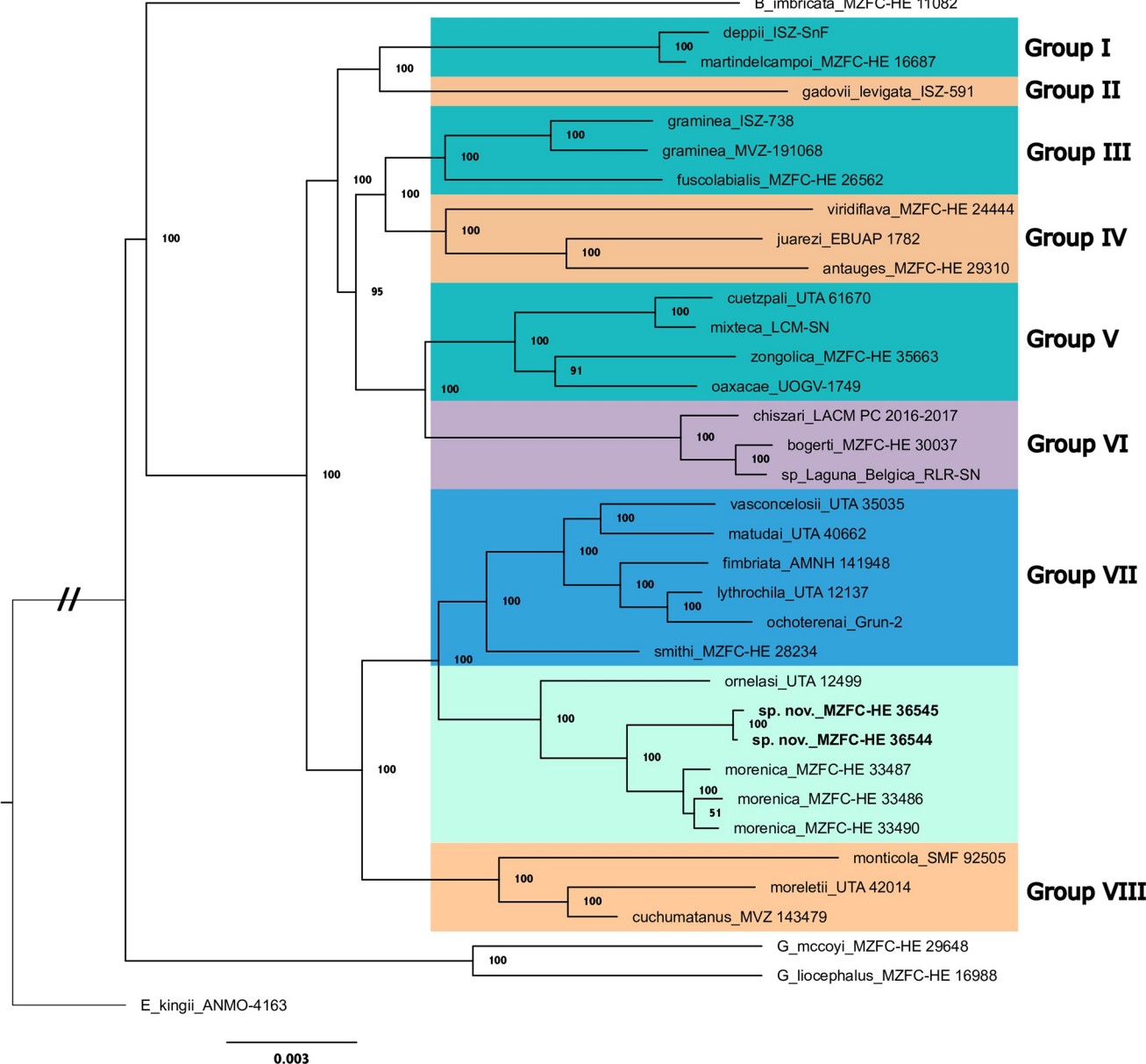

**Fig 1. Maximum likelihood phylogenetic hypothesis for the genus *Abronia* based on 3425 RADseq loci.** Numbers on branches are bootstrap values. B. = *Barisia*; E. = *Elgaria*; G. = *Gerrhonotus*. Species group names (I–VIII) follow the most recent molecular phylogenetic hypothesis of *Abronia* [38]. Color boxes indicate morphology-based taxonomic assignments [39]: subgenus *Abronia* (turquoise), subgenus *Scopaeabronia* (violet), subgenus *Auriculabronia* (blue), subgenera *Abaculabronia* plus *Lissabronia* (pale green), and former genus *Mesaspis* (pale orange).

same topology among the eight groups of *Abronia* previously defined [38]. However, the samples of the presumed new species, *A. morenica*, and *A. ornelasi* comprised a new clade within the *Abronia* tree that was the sister group to group VII [38]. The samples of the presumed new species were significantly supported as the sister taxon to *A. morenica*. The presumed new species is thus only distantly related to *A. lythrochila* (group VII, subgenus *Auriculabronia*), and is even more distantly related to *A.* sp. "Laguna Bélgica" (group VI, subgenus *Scopaeabronia*), which are the geographically closest arboreal *Abronia*. We tentatively assign the presumed new species to the subgenus *Lissabronia*, but we present alternative taxonomic arrangements in the Discussion section.

Although inconclusive regarding the distinctiveness of the presumed new species from *A. morenica*, the uncorrected pairwise genetic distances (p-distances) are informative. The mean p-distance among samples of the presumed new species and among samples of *A. morenica* were 0.000515 and 0.001123, respectively. In contrast, the mean p-distance between the presumed new species and *A. morenica* was 0.004032. This value exceeds the mean p-distances between other *Abronia* sister species that are recognized as distinct taxa (*A. deppii/A. martindelcampoi*, p-distance = 0.002076; *A. cuetzpali/A.mixteca*, 0.002684; *A. bogerti/A.chiszari*, 0.003464), yet is less than other species pairs within the genus (e.g., *A. oaxacae/A. zongolica*, 0.008179).

The conStruct cross-validation analysis of the presumed new species, *A. morenica*, and *A. ornelasi* indicated that the best model (that is, the simplest one with better predictive accuracy than others) was a model with K = 3; however, the predictive accuracy of the nonspatial and spatial models at K = 3 did not differ substantially from each other (Fig 2). This suggests that isolation by distance is not a feature of the data. Also, the analyses with both the nonspatial and spatial model and K = 3 showed that the samples of the presumed new species and the samples of *A. morenica* draw most of their ancestry from the same layer (Fig 3). However, the samples of the presumed new species also draw about one-third of their ancestry from a

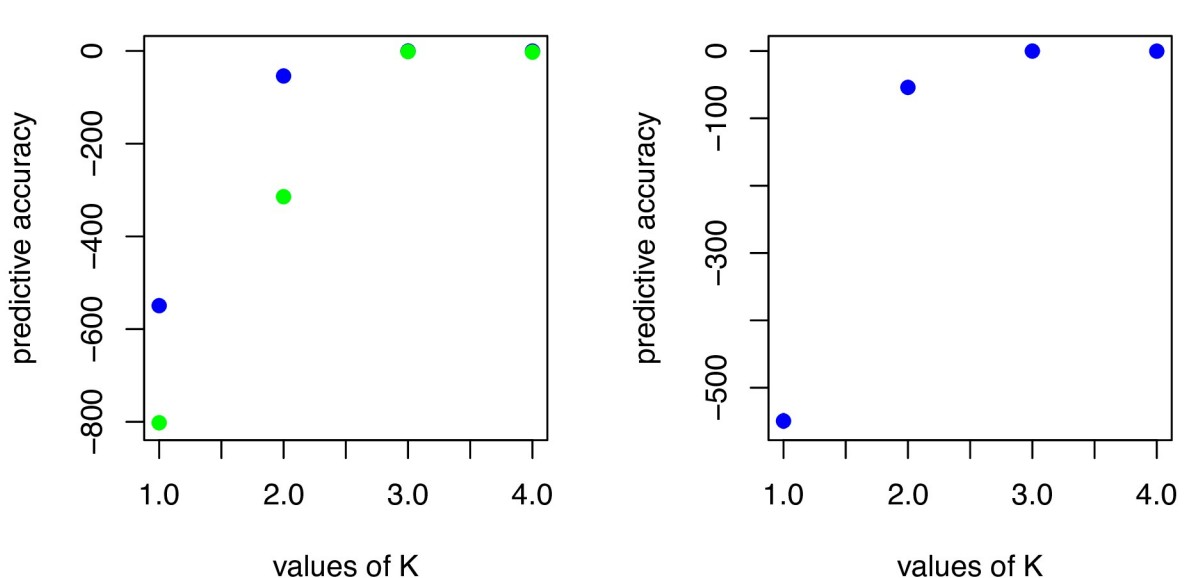

**Fig 2. Cross-validation results of conStruct analysis comparing the spatial and nonspatial conStruct models (in blue and green, respectively) run with K 1 through 4.**

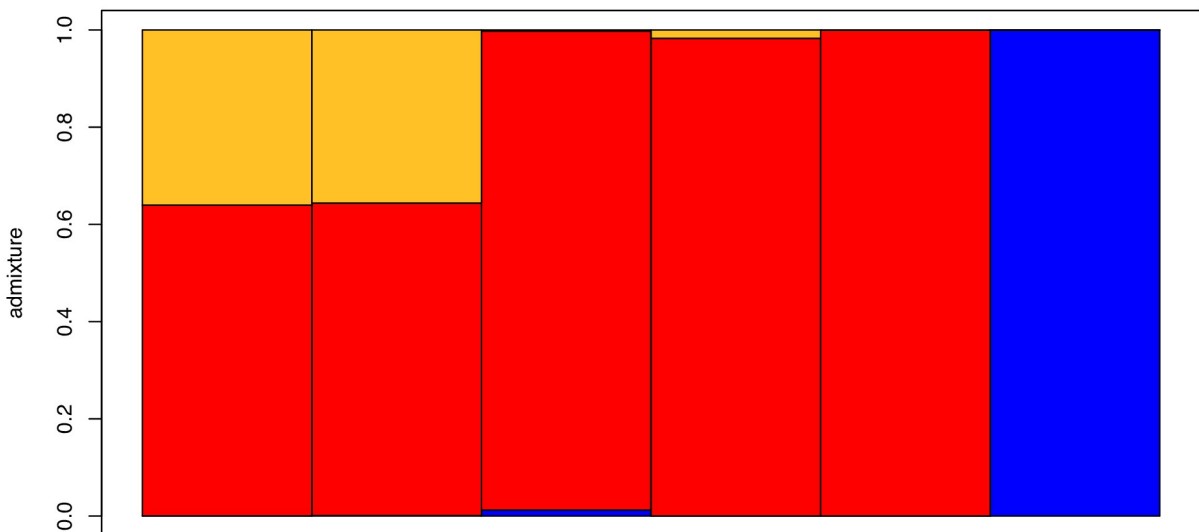

**Fig 3. Results of conStruct analysis using K = 3 for the spatial model showing layer contributions.** Samples from left to right: presumed new species (first two samples), *A. morenica* (next three samples), and *A. ornelasi*.

second layer, whereas the sample of *A. ornelasi* draws all of its ancestry from a third layer. These results are consistent with a common ancestor of the presumed new species and *A. morenica*, and they suggest moderate genetic divergence following the presumably recent geographic separation of these two lineages.

The gdi mean and 95% confidence interval (in parentheses) estimated for the presumed new species, *A. morenica*, and *A. ornelasi* were 0.8093 (0.8084–0.8101), 0.3857 (0.3847–0.3868), and 0.9246 (0.9239–0.9252), respectively (Fig 4). It has been suggested as a rule of thumb that gdi values < 0.2 suggest a single species and gdi values > 0.7 suggest distinct species, while gdi values within the range indicate ambiguous delimitation [70]. Under this interpretation framework, our BPP analysis is ambiguous regarding whether the presumed new species warrants recognition as a taxon distinct from *A. morenica*, which is congruent with the scenario of recent divergence suggested by the conStruct analysis (see above).

Morphological evidence is congruent with the genomic evidence that the presumed new species and *A. morenica* are closely related, and it is also congruent with our interpretation that they are distinct taxa. The five specimens of the presumed new species are assignable to the subgenus *Lissabronia*, to which *A. morenica* is assigned [44], because they all have 7 of the 10 diagnostic characters of that clade [1] and because they lack the diagnostic synapomorphies of any of the other recognized subgenera (*Lissabronia* uniquely lacks synapomorphies) [1, 39]. Multiple morphological features readily distinguish the presumed new species from *A. morenica* and from all other species of *Lissabronia* (see Comparisons section, below). Corroborating this morphological evidence, the presumed new species is geographically isolated from all known populations of both arboreal and terrestrial members of the genus. Within *Lissabronia*, the nearest known population is that of *A. morenica* from the type locality in the western Sierra Madre de Chiapas, some 110 km to the south-southwest [44]. The geographic gap between the presumed new species and *A. morenica* spans the lowland Central Depression of Chiapas. This broad, semi-arid depression is widely recognized as a dispersal barrier for mountain-dwelling frogs [18], salamanders [81], snakes [26, 82, 83], and lizards [27], with different species in the same genus on either side of the Depression. Based on this concordant genomic,

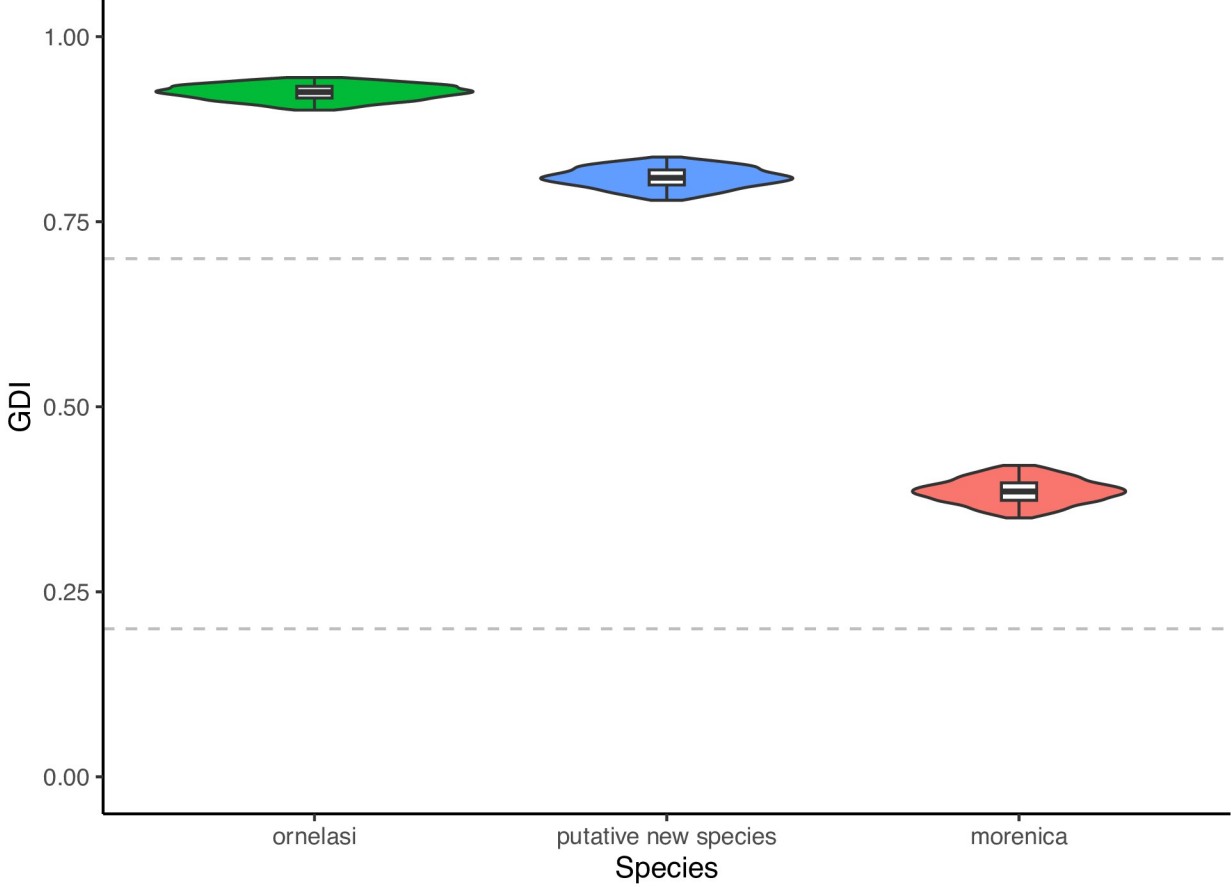

**Fig 4. Results of species delimitation in the (*A. ornelasi* (*A. morenica*, presumed new species)) clade applying the heuristic index gdi to parameter estimates from BPP.** The dotted lines correspond to gdi = 0.2 and 0.7. Horizontal lines represent means and boxes 95% confidence intervals around means.

morphological, and biogeographical evidence, we thus consider the *Abronia* samples from Coapilla to represent an undescribed species, for which we propose the name:

*Abronia cunemica* Clause, Luna-Reyes, Mendoza-Velázquez, Nieto-Montes de Oca & Solano-Zavaleta **sp. nov**.

urn:lsid:zoobank.org:act:5E56D78F-1DEF-42B3-AEBD-B2AD2E1D1817

Dragoncito de Coapilla (recommended Spanish common name)

Coapilla Arboreal Alligator Lizard (recommended English common name)

Figs 5–8, Table 1.

## Holotype (Figs 5–8, Table 1)

MZFC-HE 36544 (field series number AGC 1428), adult male, vicinity of Coapilla, Municipio de Coapilla, Northern Highlands, Chiapas, Mexico (17.14˚, -93.16˚, datum WGS 84), 1625 m elevation, Adam G. Clause, Emmanuel Javier-Vázquez, Ana Reyna Pale Morales, 14 August 2021.

## Paratypes (Figs 6 and 7, Table 1, n = 4)

MZFC-HE 36545 (field series number AGC 1429), adult female, all collection data the same as for the holotype. MZFZ 4512 (AGC 1484), juvenile male, vicinity of Coapilla, Municipio de

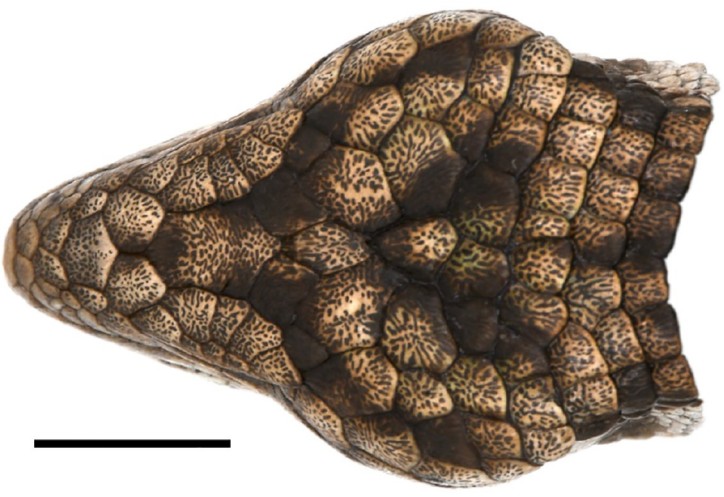

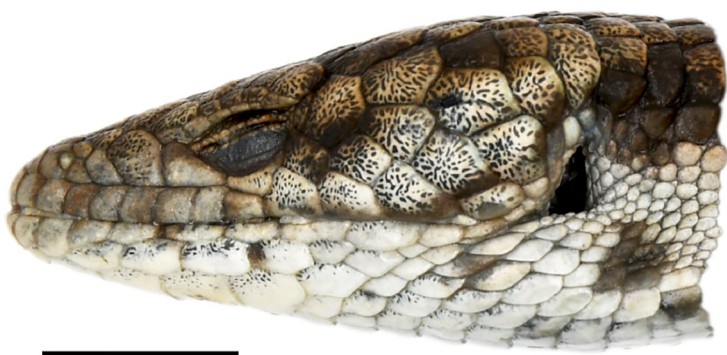

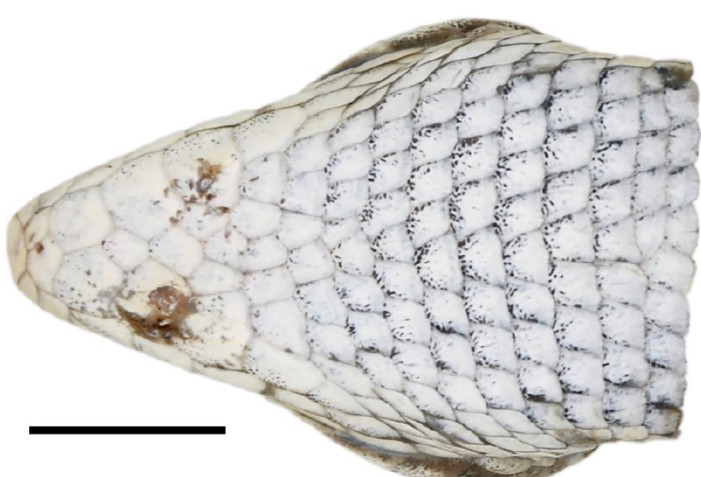

**Fig 5. Holotype of *Abronia cunemica* sp. nov. from Coapilla, Chiapas, Mexico (MZFC-HE 36544, 29 mm head length).** Dorsal view (top), left lateral view (middle), and ventral view (bottom) of head in preservative. All scale bars = 10 mm. Photographs by Israel Solano-Zavaleta.

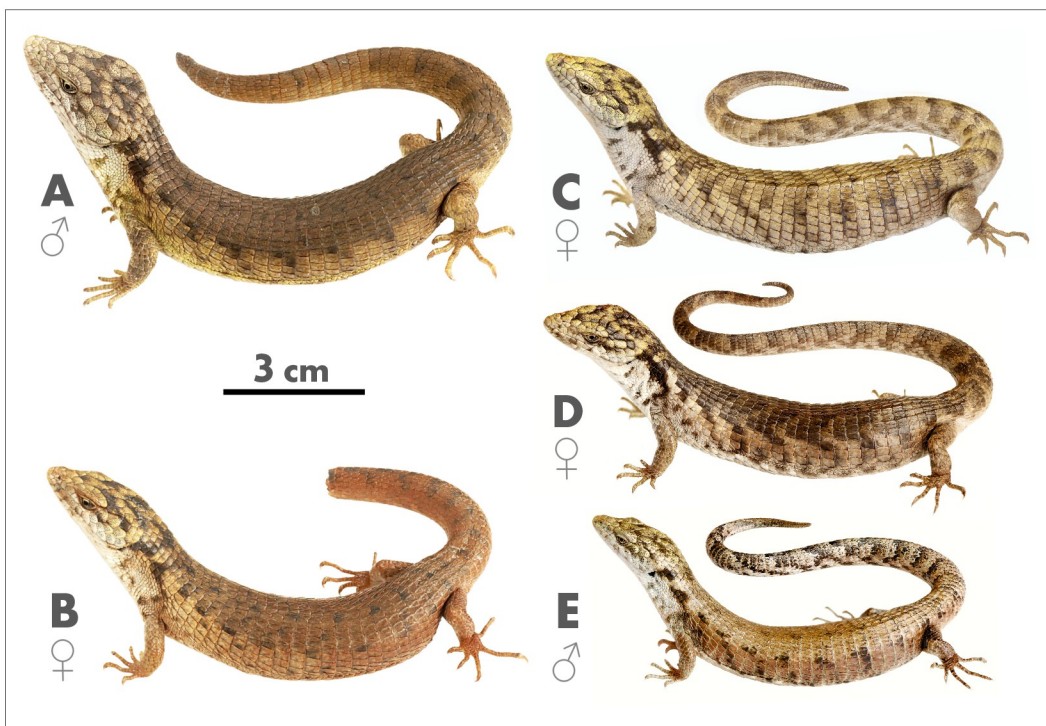

**Fig 6. Color variation in life of type series of *Abronia cunemica* sp. nov. from Coapilla, Chiapas, Mexico.** (A) Adult male holotype, MZFC-HE 36544, 127 mm snout-to-vent length (SVL); (B) adult female paratype, MZFC-HE 36545, 113 mm SVL; (C) adult female paratype, MZFZ 4514 (AGC 1492), 110 mm SVL; (D) adult female paratype, MZFZ 4513 (AGC 1491), 107 mm SVL; (E) juvenile male paratype, MZFZ 4512 (AGC 1484), 91 mm SVL. Photographs by Emmanuel Javier-Vázquez.

Coapilla, Northern Highlands, Chiapas, Mexico (17.13˚, -93.16˚, datum WGS 84), 1605 m elevation, Oscar M. Mendoza-Velázquez, Candelario Cundapí-Pérez, Roberto Luna-Reyes, Adam G. Clause, Marcos Joaquín Fitz-Pérez, José Manuel Toledo-Morales, Emmanuel Javier-Vázquez, Daniel Lara-Tufiño, 16 February 2022. MZFZ 4513–4514 (AGC 1491–1492), two adult females, all collection data the same as for MZFZ 4512 except collected on 18 and 19 February 2022, respectively. The GPS coordinates in this paragraph and the preceding paragraph are intentionally imprecise; details are available in the Conservation subsection.

## Diagnosis

*Abronia cunemica* sp. nov. can be distinguished from all recognized congeners, including all species formerly considered members of the genus *Mesaspis*, by the following combination of characters: (1) lack of protuberant or spine-like supra-auricular scales; (2) lack of protuberant or casque-like posterolateral head scales; (3) dorsum of head pale yellow with distinct dark markings; (4) 35–39 transverse dorsal scale rows; (5) lateralmost row of ventral scales enlarged relative to adjacent medial row; and (6) dorsum brown with dark crossbands, these sometimes reduced to series of dark spots.

## Comparisons

*Abronia cunemica* sp. nov. can be differentiated from all species of the former genus *Mesaspis* by having 35–39 transverse dorsal scale rows (vs. > 40 rows). Additionally, the new species

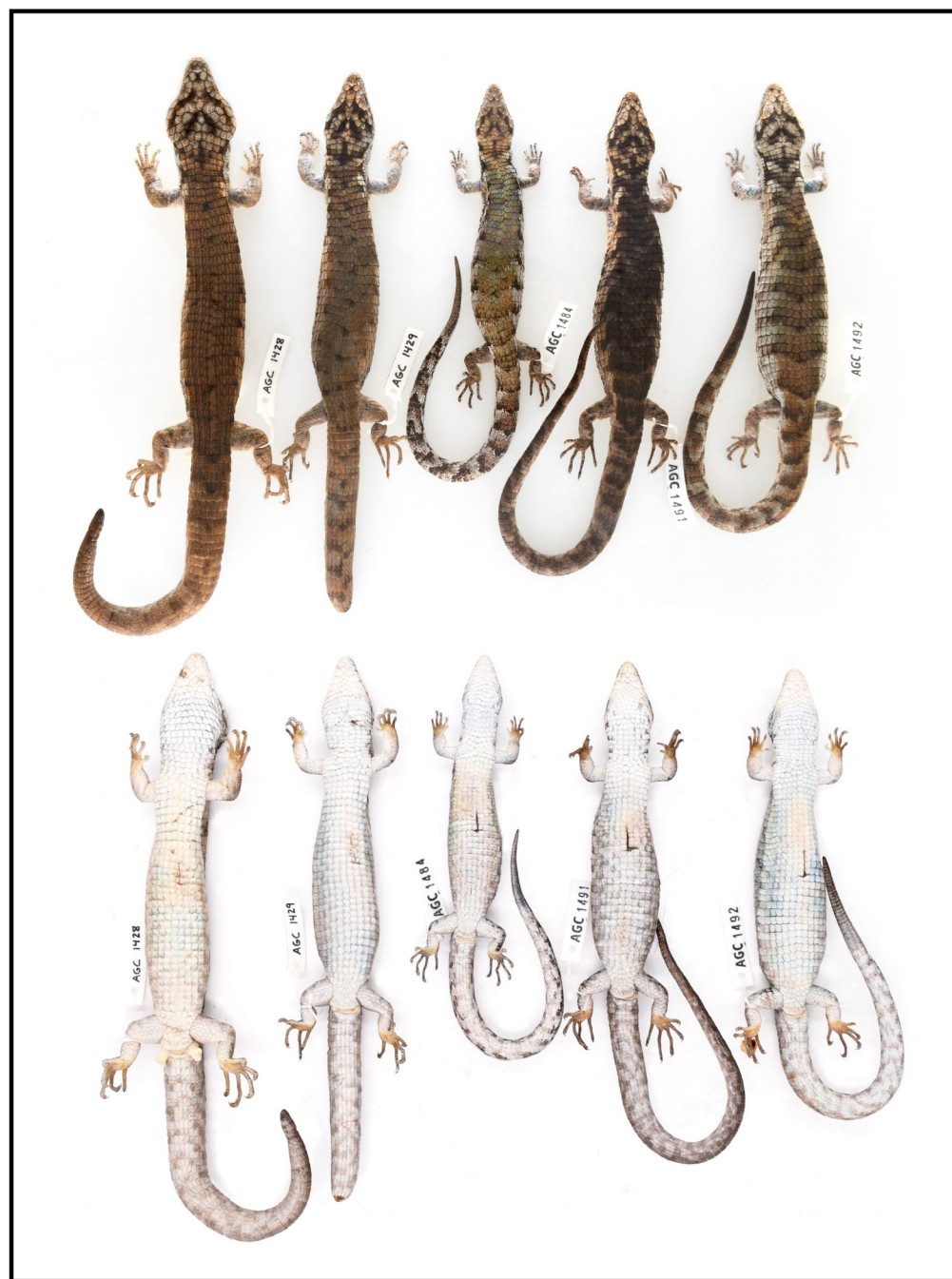

**Fig 7. Color variation in preservative (ethanol after formalin) in dorsal and ventral view of type series of *Abronia cunemica* sp. nov. from Coapilla, Chiapas, Mexico.** From left to right: adult male holotype, MZFC-HE 36544 (AGC 1428), 127 mm snout-to-vent length (SVL); adult female paratype, MZFC-HE 36545 (AGC 1429), 113 mm SVL; juvenile male paratype, MZFZ 4512 (AGC 1484), 91 mm SVL; adult female paratype, MZFZ 4513 (AGC 1491), 107 mm SVL; and adult female paratype, MZFZ 4514 (AGC 1492), 110 mm SVL. Photographs by Israel Solano-Zavaleta.

differs from *A. cuchumatanus*, *A. gadovii*, and the *A. moreletii* species complex by having 14 longitudinal dorsal scale rows (vs. 16 or 18, 16–18, and 18–22, respectively); from *A. antauges* and *A. juarezi* by having vertebral and paravertebral dorsal scales strongly keeled at midbody (vs. smooth to slightly convex); from *A. viridiflava* by having the frontonasal scale present (vs.

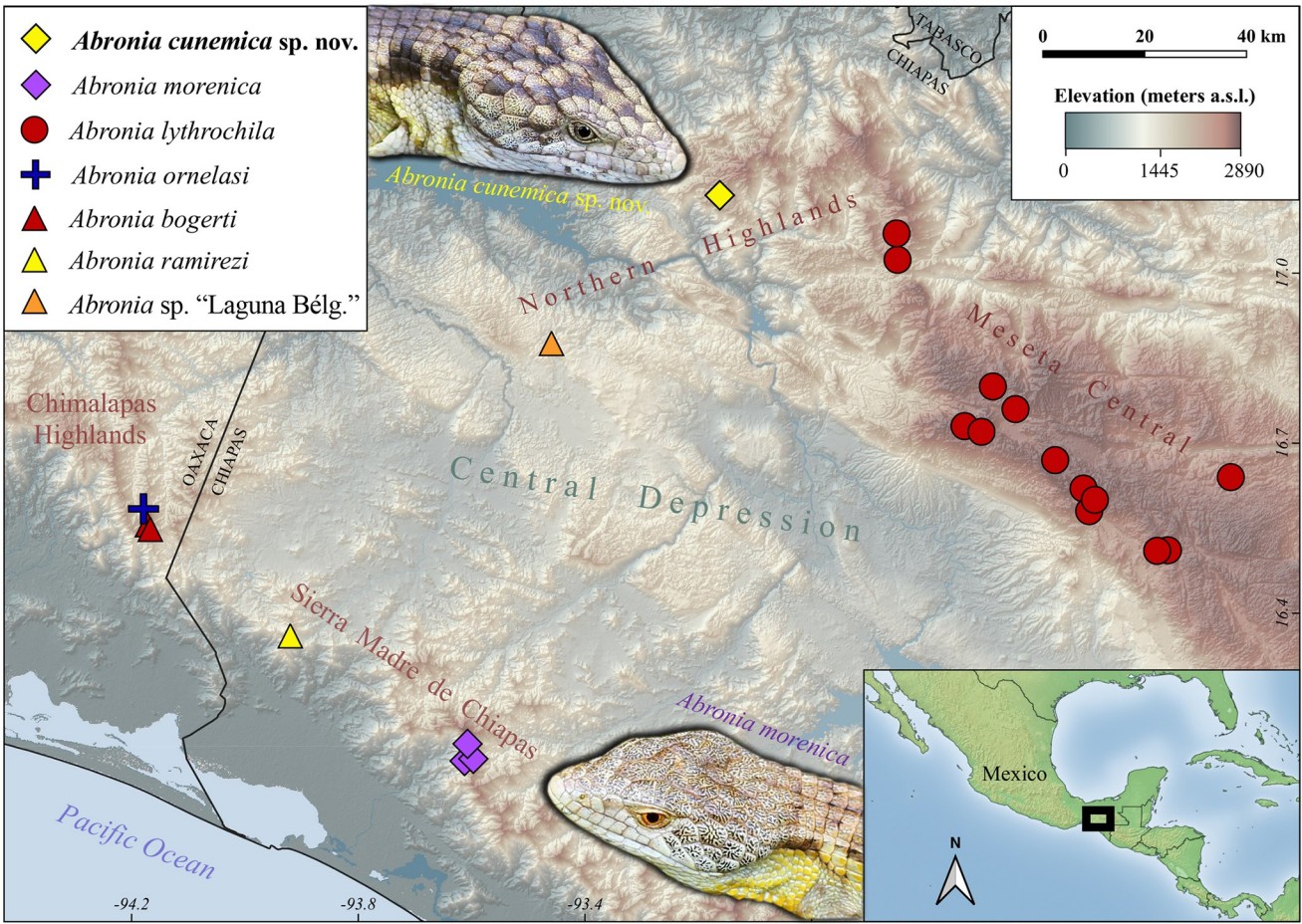

**Fig 8. Geographic distribution of *Abronia cunemica* sp. nov. and nearby arboreal congeners in Chiapas and Oaxaca, Mexico.** Inset photographs show characteristic differences in adult dorsal head color between *A. cunemica* sp. nov. (pale yellow, with dark markings distinct) and the closely related *A. morenica* (pale gray/tan, with dark markings absent or faint). The holotypes (adult males) of both species are shown; photograph of *A. cunemica* sp. nov. intentionally mirrored horizontally. Photographs by Adam G. Clause. Map layers courtesy of Natural Earth (public domain).

absent); and from *A. monticola* by having a divided or partly divided postmental scale in 3/5 or 60% of specimens (vs. undivided). Among *Abronia* species not previously considered members of the former genus *Mesaspis*, *Abronia cunemica* sp. nov. differs from each recognized subgenus as follows, with character state(s) for each subgenus in parentheses and subgeneric synapomorphies (if they exist) indicated in italic font. Unlike the subgenus *Scopaeabronia*, the new species has the lower primary temporal scale unexpanded (vs. *expanded*), 6 longitudinal nuchal scale rows (vs. *8 rows*) and 35–39 transverse dorsal scale rows (vs. *38–47*). Unlike the subgenus *Auriculabronia*, the new species lacks strongly protuberant or spine-like supra-auricular scales (vs. *present*). Within the subgenus *Auriculabronia*, the new species further differs from *A. matudai* (a species in which the supra-auricular scales are sometimes barely protuberant) in having a dark lateral bar on the neck present, albeit often divided into two separate blotches (vs. absent); and in having a dorsum that is brown with dark crossbands, albeit often reduced to series of spots (vs. dorsum green with no crossbands in adult males). Unlike the subgenus *Abronia*, the new species has the lateralmost row of ventral scales distinctly enlarged relative to the adjacent medial row (vs. not enlarged). Within the subgenus *Abronia*, the new species further differs from the *deppii* group (*A. cuetzpali*, *A. deppii*, *A. martindelcampoi*, *A.*

**Table 1. Selected characteristics of *Abronia cunemica* sp. nov. and all species in the *Lissabronia* congeneric clade.**

| Character | *A. cunemica* sp. nov. | *A. frosti* | *A. montecristoi* | *A. morenica* | *A. salvadorensis* |
|---|---|---|---|---|---|
| Adult dorsal body color-pattern in life | Brown with dark crossbands (sometimes indistinct) | Black or blackish brown with white or yellow transverse markings | Brown sometimes with pale crossbands | Brown with dark crossbands (sometimes indistinct) | Pale brown with dark crossbands |
| Adult dorsal head color-pattern in life | Pale yellow with prominent dark markings | Black with whitish, yellowish, or pale gray markings | Brown or gray with dark markings absent | Pale gray or tan with dark markings absent or faint | Brown or gray with dark markings present or absent |
| Dark lateral bar on neck from shoulder to ear opening | Present (often broken into two blotches) | Absent | Absent | Present | Present |
| Yellow or orange spots on flanks | Absent (but flanks sometimes mostly pale yellow) | Absent | Absent | Present | Absent |
| Primary temporal scales | 4 | 2 | 4 | 4 | 4 |
| Occipital scales* | 1 | 1 | 3† | 1 (86%) or 3 (14%) | 1 (25%) or 3 (75%)†† |
| Transverse dorsal scale rows | 35–39 | 28–32 | 30–31 | 30–35 | 29–32 |
| Longitudinal ventral scale rows | 12 | 14–16 | 12 | 12 | 12–14 |
| Adult snout-to-vent length (mm) | 107–127 | 100–110 | 85–93††† | 92–93 | 80–111 |

* = percentages represent the proportion of available specimens with the given number of occipital scales.

† = count follows interpretation by Campbell et al. [1]; all previous authors counted five scales.

†† = count follows interpretation by Campbell and Frost [39] and Clause et al. [44].

††† = Hidalgo [120] gives the snout-to-vent length of the largest known specimen (KU 184046) as 90 mm, but we here follow the reported measurement from Campbell and Frost [39].

*mixteca*, and *A. oaxacae* as defined by Campbell et al. [84], contra Campbell and Frost [39]) in having dorsal scales on the flanks arranged in parallel longitudinal rows relative to the ventro-lateral fold (vs. *oblique longitudinal rows*), and from *A. zongolica* by having 35–39 transverse dorsal scale rows (vs. 30–34). Unlike the subgenus *Aenigmabronia*, the new species has one occipital scale (vs. *two occipitals*), and two scale rows separating the occipital scale from the first transverse nuchal scale row (vs. *one scale row separating the occipitals from the nuchals*). Unlike the subgenus *Abaculabronia*, the new species has a brown dorsum with at least traces of crossbands in life in adults (vs. *dorsum olive green with pale or yellow scale margins and no trace of crossbands in life in adults*), and supranasal scales lacking contact in all specimens (vs. *supranasals in contact in 8/9 or 89% of specimens*). Within the subgenus *Lissabronia*, no single character differentiates the new species from all four currently recognized members of the group. However, it can be distinguished from all *Lissabronia* species except for *A. morenica* by having 35–39 transverse dorsal scale rows (vs. 28–32). Unlike *A. morenica*, the new species attains a larger adult size of 107–127 mm snout-to-vent length (vs. 92–93 mm), and the dorsum of the head is pale yellow with distinct dark markings (vs. pale gray or tan with dark markings absent or faint). Table 1 presents additional characters that differentiate species of *Lissabronia* from *A. cunemica* sp. nov. and/or from each other.

We are uncertain about the subgenus to which *A. cunemica* sp. nov. belongs. Nonetheless, we provisionally assign it to *Lissabronia* based on our phylogenetic analysis (see above). Furthermore, unlike for all other subgenera, no single character cleanly separates the new species from *Lissabronia* based on the diagnosis provided by Campbell et al. [1] and on the subsequent description of *A. morenica* [44] that provisionally assigned that species to *Lissabronia*. In the

Discussion section we present further explanation for our tentative assignment of the new species.

## Description of holotype (Figs 5–8, Table 1)

Adult male with both hemipenes everted, mass 37.6 g, snout-to-vent length (SVL) 127 mm, head length from rostral to upper anterior edge of ear opening 29 mm, head width at broadest point 22 mm, head width/length ratio = 75.9%, tail broken and regenerating, tail length 121 mm, and 50 caudal whorls (including regenerating portion).

Supranasals 1/1, neither expanded medially; postnasals 2/2, upper smaller than lower; one pair each of anterior and posterior internasals situated between rostral and frontonasal, right anterior internasal abnormally divided into a large medial and a small lateral scale; prefrontals >2 times size of posterior internasals, broadly contacting each other medially; canthals 1/1, separating posterior internasal and prefrontal; loreals 1/1, in contact with both postnasals; cantholoreals 1/1, barely extending onto dorsum of canthus rostralis, narrowly contacting anterior median supraocular, broadly contacting canthal, prefrontal, and supralabials; median supraoculars 5/5; lateral supraoculars 3/3; superciliaries 6/6, anteriormost contacting cantho-loreal and <1.5 times length of adjoining superciliary; preoculars 1/1, left fused with anterior subocular; suboculars 1/2, anterior on left aberrantly fused with preocular; posterior subocu-lars not contacting lowermost primary temporal; postoculars 2/3; frontal large, not contacting frontonasal, narrowly contacting interparietal; parietals lacking contact with median suprao-culars; one large occipital, slightly larger than interparietal; two transverse scale rows separat-ing occipital from first transverse row of nuchals; primary temporals 4/4, only lowermost two contacting postoculars on each side, third aberrantly fused with fourth secondary temporal on each side; secondary temporals 4/4, fourth aberrantly fused with third primary temporal on each side; tertiary temporals 4/5; supralabials 11/10, antepenultimate posteriormost to reach orbit; infralabials 8/8; postmental divided; three pairs of enlarged chin shields posterior to divided postmental, posteriormost ones subequal in size to adjacent chin shields, > 1.5 times size of posterior scales; sublabials 5/5, anteriormost contacting second infralabial but not postmental.

Minimum longitudinal nuchal scale rows 6; transverse dorsal scale rows 36–38; longitudinal dorsal scale rows 14, arranged in parallel horizontal rows on sides of body; eight middorsal longitudinal scale rows strongly keeled, becoming smooth on flanks; transverse ventral scale rows 37; longitudinal ventral scale rows 12; lateralmost row of ventral scales enlarged relative to adjacent medial row, but only on middle third of body; osteoderms appear moderately well developed on head and adjacent nuchals; posterolateral corners of head not protruding or cas-que-like; osteoderms appear weakly developed or absent on body and tail; supra-auricular scales granular, not protruding or spine-like; scales on side of neck between enlarged lateral nuchals and enlarged ventrolateral scales (hereafter, lateral neck scales) 8–10, granular; ante-brachials from insertion of forelimb to wrist 12–13; ventrolateral fold between ear opening and forelimb absent; ventrolateral fold posterior to forelimb moderately well developed, with 3–6 indistinct longitudinal rows of small scales and granules on interstitial skin; subdigital lamellae on fourth toes 19/19.

## Coloration of holotype

In life, body medium ochre brown becoming pale yellow on lateral nuchals and dull lemon yel-low with brown flecks on flanks. Body with distinct dark brown flank blotches and faint dark brown middorsal spots, seemingly the remnants of 10 indistinct crossbands. Dark blotches and spots separated by 3 scales middorsally and 1.5–2.5 scales laterally. Most scales in and

along ventrolateral fold dull lemon yellow, but some mostly or entirely reddish brown due to ventrolateral continuation of crossbands. Lateral neck scales mostly whitish to dull lemon yellow, with 3–4 distinct reddish-brown ventrolateral blotches. Lateral neck scales bordered dorsally by prominent, interrupted or nearly interrupted black to dark brown bar extending from shoulder to near upper posterior edge of ear opening, 1–2 times width of lowermost nuchal scales. Dorsal surface of forelimbs and hindlimbs rusty brown, with a few yellow highlights and dark brown flecking and blotching, more so on forelimbs. Forelimb digits dull yellow with rusty brown spotting, hindlimb digits rusty brown. Head pale yellow becoming pale gray to white laterally with prominent, ill-defined arrowhead-shaped black to dark brown dorsal blotches. Head scales with moderate rugosity accentuated by fine black to dark brown flecking and vermiculations that extend onto most nuchals. Scales between nares on dorsum, and between orbit and rostral on sides of head (except cantholoreal), with no dark flecking or vermiculations. Tail same color as body, with 11 indistinct but mostly uninterrupted dark brown to blackish crossbands that degrade into a checkerboard-like pattern ventrally. Lower jaw, chin, and throat white to pale gray, many scales with distinct dark flecking on anterior or dorsal margins, especially prominent on sublabials and chinshields. Venter anterior to forelimbs white to pale gray, becoming yellow on remainder of venter including hindlimbs and base of tail, and beige with faint yellow tinge on remainder of tail. Many ventral scales with prominent pale ochre brown markings, becoming more extensive laterally. Manus and pes rusty orange to rusty yellow. Iris pale yellow with heavy dark flecking.

In preservative (ethanol after formalin), body medium ochre brown with small parts of some scales becoming greenish, cream on lateral nuchals and horn-colored with brown flecks on flanks. Body with distinct brown flank blotches and faint brown dorsal spots. Most scales in and along ventrolateral fold horn-colored, but some mostly or entirely brown due to ventrolateral continuation of crossbands. Lateral neck scales mostly whitish to horn-colored, with 3–4 distinct greenish brown ventrolateral blotches. Lateral neck scales bordered dorsally by prominent, interrupted or nearly interrupted dark brown bar extending from shoulder to near upper posterior edge of ear opening. Dorsal surface of forelimbs and hindlimbs olive, with a few cream highlights and light brown flecking and blotching, more so on forelimbs. Fingers horn-colored to whitish with light brown spotting, toes light brown to olive. Head pale gray becoming white laterally with prominent, vaguely arrowhead-shaped brown dorsal blotches. Head scales with moderate rugosity accentuated by fine brown flecking and vermiculations that extend onto most nuchals. Tail lighter than body, with mostly light brown to brown crossbands that fade to horn-colored ventrally. Lower jaw, chin, and throat white to cream. Venter anterior to forelimbs white, becoming mostly cream and horn-colored laterally over the remainder of venter including limbs and base of tail, and grayish-cream on remainder of tail. Manus and pes saffron.

## Variation (Figs 6 and 7)

All four paratypes are similar to the holotype in most respects, but they differ as follows. Three adult females (MZFC-HE 36545 and MZFZ 4513–4514) with mass 22.1–24.3 g ($\bar{X}$ = 23.4), SVL 107–113 mm ($\bar{X}$ = 110), head length from rostral to upper anterior edge of ear opening 22–23 mm ($\bar{X}$ = 22.3), head width at broadest point 16–17 mm ($\bar{X}$ = 16.3), head width/length ratio 72.7–73.9% ($\bar{X}$ = 73.1%), tail unbroken and unregenerated only in MZFC-HE 36545, with tail length 156 mm (1.46 times SVL) and 97 caudal whorls. One juvenile male (MZFZ 4512) with mass 13.9 g, SVL 91 mm, head length 18 mm, head width 12 mm, head width/length ratio 66.7%, tail broken and regenerating, tail length 126 mm (including 26 mm regenerated tissue).

Left supranasal expanded medially and displacing adjacent posterior internasal in MZFC-HE 36545, and left supranasal damaged in MZFZ 4514; medial azygous scale present between anterior and posterior internasals in MZFZ 4514; prefrontals 2 times size of posterior internasals in MZFZ 4512, 2.5 times size of posterior internasals in MZFC-HE 36545 and MZFZ 4513–4514, but not contacting each other in MZFC-HE 36545 and narrowly contacting each other in MZFZ 4513; cantholoreal not (left) or barely (right) contacting anterior median supraocular in MZFC-HE 36545, no median supraocular contact in MZFZ 4513, and very narrow contact in MZFZ 4514; superciliaries 7/7 in MZFC-HE 36545, MZFZ 4512, and MZFZ 4513; anteriormost superciliary nearly 2 times length of adjoining superciliary in MZFC-HE 36545 and MZFZ 4513, and nearly same length as adjoining superciliary in MZFZ 4512; suboculars 2/3 in MZFZ 4514, right anteriormost aberrantly divided into small posterior and large anterior scales, and 2/2 on all other paratypes; posterior suboculars not contacting lowermost primary temporal in all paratypes; postoculars 4/4 in MZFC-HE 36545 and MZFZ 4514, 3/3 in MZFZ 4512, and 3/4 in MZFZ 4513; frontal narrowly contacting frontonasal in MZFC-HE 36545, barely contacting frontonasal via thin spur in MZFZ 4512 and MZFZ 4514; occipital smaller than interparietal in all paratypes, abnormally divided posteriorly to form a small semi-triangular scale on left side in MZFC-HE 36545 and right side in MZFZ 4513; two transverse scale rows separating occipital from nuchals with enlarged posteriormost scale medially dividing first row of nuchals in MZFZ 4513; third primary temporal aberrantly fused with uppermost secondary temporal on each side only in MZFC-HE 36545; secondary temporals 3/4 in MZFZ 4512 and MZFZ 4513, second on left and third on right abnormally enlarged posteriorly and partially displacing adjacent tertiary temporals in MZFZ 4512, first and second on left separated by abnormally enlarged second tertiary temporal in MZFZ 4513, second on left abnormally small in MZFZ 4514; tertiary temporals 4–5 ($\bar{X} = 4.6$), second abnormally enlarged and anteriorly displaced in MZFZ 4513; supralabials 10–11 ($\bar{X} = 10.3$); infralabials 8–9 (23345076), small aberrantly divided sublabial nearly separating eighth and ninth sublabials on right side in MZFZ 4514; postmental single in MZFZ 4513, partly divided posteriorly in MZFC-HE 36545 and anteriorly in MZFZ 4514; second sublabial on left abnormally divided in MZFZ 4513, fifth sublabial on both sides aberrantly divided and anteriormost contacting second (left) or third (right) infralabial in MZFZ 4514, sublabials 4/5 with anteriormost contacting third (left) or second (right) infralabial in MZFZ 4512.

Transverse dorsal scale rows 35–39 ($\bar{X} = 37.25$); eight middorsal longitudinal scale rows strongly to moderately keeled in MZFZ 4512 and MZFZ 4514; transverse ventral scale rows 37–40 ($\bar{X} = 38.5$); lateral neck scales 8–9 in MZFC-HE 36545, 9–10 in MZFZ 4513–4514; subdigital lamellae on fourth toes 18–21 ($\bar{X} = 19.6$).

In life, the three adult female paratypes (MZFC-HE 36545 and MZFZ 4513–4514) differ from the adult male holotype as follows: body dark brown with rusty tinge (MZFZ 4513) to pale ochre brown (MZFZ 4514); flanks medium rusty brown (MZFZ 4513) to pale brown (MZFZ 4514), becoming pale gray, beige, or cream anteriorly; remnants of dark crossbands even more reduced on MZFC-HE 36545, but crossbands unbroken, distinct, and roughly chevron-shaped on MZFZ 4513–4514, separated by 0.5–2 scales middorsally and 1.5–3.5 scales laterally; scales in and along ventrolateral fold same color as adjacent flank scales, including ventrolateral continuation of darker crossbands; lateral neck scales whitish to pale gray, rarely with yellow tinge, and ventrolateral blotches blackish to medium brown; dark lateral neck bar interrupted in MZFC-HE 36545, complete in MZFZ 4513–4514; dorsal surface of forelimbs and hindlimbs whitish, pale gray, pale brown, medium brown, or rusty brown, often with darker flecks or blotches and with forelimbs always at least partially paler than hindlimbs; digits various shades of gray or brown, often with darker spots or markings, rarely with cream

flecks or highlights; 4–5 alternating brown spots on supralabial and/or infralabial scales in MZFC-HE 36545 and MZFZ 4513; cantholoreal scales lack dark flecks and vermiculations; tail with 9 (MZFC-HE 36545, tail broken and not yet regenerating), 14 (MZFZ 4514, tail regenerating), or 22 (MZFZ 4513, tail unbroken) dark brown crossbands, some partially merged with adjacent crossbands; regenerated portion of tail in MZFZ 4514 almost completely dark brown; lower jaw, chin, and throat lack dark flecking; venter white to pale gray becoming darker gray laterally, with pale yellow or cream midventral stripe posterior to forelimbs; venter of tail white to pale gray with degraded crossbands forming ill-defined checkerboard-like pattern of darker gray blotches; manus and pes bright rusty orange to dull rusty brown, sometimes becoming yellowish basally; iris with faint green tinge in MZFC-HE 36545.

In life, the juvenile male paratype (MZFZ 4512) differs from the adult male holotype as follows: body pale brown becoming paler beige on flanks, with some darker flecking throughout and a faint cream tinge anteriorly; body with 11 indistinct dark brown crossbands, largely reduced to blotches on vertebral and paravertebral scale rows, often bordered posteriorly with pale gray to beige flecks or spots, each separated by 2–3 scales both middorsally and laterally; most scales in and along ventrolateral fold beige, but some mostly dark brown bordered posteriorly by white or pale gray spots due to ventrolateral continuation of crossbands; lateral neck scales whitish, some with cream tinge, and ventrolateral blotches medium brown; dark lateral neck bar interrupted; forelimbs and digits beige to pale brown with medium brown flecks and spots; hindlimbs and digits medium brown with dark brown flecks and spots; 3–4 alternating pale brown spots on supralabial scales; pale yellow and dark brown markings on dorsum of head faded and less contrasting; head scales smooth or weakly rugose, with less extensive dark flecking and vermiculations; tail slightly paler than body, especially posteriorly, but dark crossbands more distinct than on body; regenerated portion of tail pale brown with darker flecking and no crossbands; lower jaw, chin, and throat lack dark flecking; venter anterior to forelimbs white to pale gray, becoming pale yellow on remainder of venter including limbs and base of tail, and beige to pale gray on remainder of tail; manus and pes dull yellow-orange.

In preservative (ethanol after formalin), all paratypes maintain their basic color pattern elements from life except that rusty brown tones are mostly replaced by various shades of tan, cream, or pale gray; dorsum of body dark brown to pale gray or pale brown, but scales in lighter-colored paratypes with greenish tones; dark dorsal crossbands slightly faded and less distinct; neck generally more grayish, tan, or pale brown; dorsum of head grayish; head markings accentuated and distinct in all specimens; faint to distinct oblique dark mark extends from posterior margin of frontal to tertiary temporals in all specimens; manus and pes yellowish to saffron.

## Etymology

The species name is a feminine singular adjective in the nominative case derived from Cuñemo (alternative spellings: Kuñømø or Kujnyä'mä), which is the name for Coapilla in the indigenous Zoque language. Coapilla is derived from the Náhuatl words *coatl* (snake) and *apan* (river) and means "river of the snakes," while Cuñemo has been variously translated as *agua entre los árboles* ("water among the trees" in English) [85], *lugar de la gran capital* ("place of the new or great capital") [86], or *corona de cerros* ("crown of hills") according to residents of the area. The chosen name, derived from the Zoque language, thus refers to the ejido and municipality which support the only known population of the new species. Our inquiries with residents of Coapilla about this name received universally positive responses. The recommended English pronunciation is "koon-YEM-ih-kuh."

## Distribution and ecology

Like most arboreal *Abronia* species described in the last 30 years, *A. cunemica* sp. nov. has only been reported from the vicinity of the type locality (Fig 8). However, also like many other members of the genus, this species probably occurs more widely given the presence of seemingly suitable interconnected highland forests. All documented populations exist on a small, wooded tableland. To the north and east, this tableland transitions into steep slopes that form several peaks exceeding 2200 m elevation which overlook the town of Tapalapa. To the south and west, the tableland ends sharply in a long bank of dramatic cliffs and promontories that loom above the town of Copainalá. Rainfall on this tableland drains either into the Río Zacalapa to the west or the Río Chavarria to the east. Both waterways then feed into the Gulf of Mexico, as part of the Río Grijalva basin.

The woodland in and around the type locality for *A. cunemica* sp. nov. is pine-oak forest, and includes *Pinus chiapensis*, *P. maximinoi*, *Quercus crispipilis*, and *Q. peduncularis* (Fig 9B). Although uncommon, *Carpinus caroliniana* and *Liquidambar styraciflua* trees are also present. Epiphytic growth in occupied forest is variable (Fig 9A and 9C). In some patches it is extensive, dominated by dense masses of *Tillandsia fasciculata* and/or *T. rodrigueziana*. Other, less abundant bromeliads include *T. juncea*, *T. comitanensis* and/or *T. makoyana*, and an unidentified species of *Catopsis*. Many trees support *Toxicodendron radicans* vines, which at times form large tangles that ascend high into the canopy. Less conspicuous elements of the epiphytic flora include at least one species of mistletoe, a few species of ferns, and several orchids that include *Domingoa purpurea* and one or more species in the genera *Epidendrum* or *Prosthechea*. Epiphytic moss is generally scarce or absent, but branches of many trees are laden with crustose, foliose, and fruticose lichens in the genera *Cladonia*, *Leptogium*, *Parmotrema*, *Pseudocyphellaria*, and *Usnea* [87].

On five separate trips to the vicinity of Coapilla (22–23 August 2015, 9–10 August 2021, 13–15 August 2021, 15–19 February 2022, and 5–7 August 2022), we cumulatively invested over 350 person-hours of search effort specifically for *A. cunemica* sp. nov. in occupied habitat, including climbing with ropes into the canopy of almost 20 trees to search among the epiphytes. Despite this effort, we found only five individuals. This difficulty in encountering the species was echoed in our conversations with local residents. Of approximately two dozen people that we met while walking trails and roads through occupied forest, only about half of them recognized the *Abronia* when shown a photograph, video, or live animal. Furthermore, a student who completed extensive herpetological fieldwork in the Municipality of Coapilla, including one site that overlaps the type locality of *A. cunemica* sp. nov., reported no observations of the species [88].

The arboreal behavior of *A. cunemica* sp. nov., together with the dense epiphytic growth in many trees of the region, are likely responsible for this lack of familiarity and few observations. We found three specimens 3.5–19.5 m high on limbs of both live and dead *Pinus chiapensis* trees. We also found a pair in a courtship bite-hold on the forest floor, after they presumably fell from an adjacent *Quercus* sp. tree. We observed individuals from 10:30–13:30 hr, during sunny or partly cloudy conditions with ambient air temperatures of 20–27˚C. Consistent with the summer/fall mating season documented in other *Abronia* species [89–91], we found the courting pair of *A. cunemica* sp. nov. on 14 August. Although the male immediately released his bite hold on the female upon capture, a few hours later when returned to close contact with her, he re-initiated a more persistent bite hold on her head and neck. After being manually separated, a few minutes later we again brought the pair into close contact. The male responded with repeated bouts of subtle yet intense tail tremors lasting several seconds. Two adult females subsequently collected in mid-February appeared to be gravid, although we did not dissect the specimens to confirm this suspicion.

The male holotype of *Abronia cunemica* sp nov. is unusually large, measuring 127 mm SVL. Among arboreal Mexican species of the genus, only *A. mixteca* is reported to be larger, with a maximum known SVL of 148 mm [39]. The Guatemalan endemic species *A. anzuetoi* and *A. fimbriata* are the only other species known to exceed *A. cunemica* sp. nov. in size, attaining lengths of 143 and 130 mm SVL, respectively [39, 92]. However, we are aware of unpublished records of *A. lythrochila* exceeding 127 mm SVL, and we hope to see this data formally announced soon.

Remarkably, one adult female paratype (MZFZ 4514) of *A. cunemica* sp. nov. was re-found and collected 97 days after she was captured, photographed, and released by OMMV on 14 September 2021. Her location when re-captured was less than 10 m from her original capture location, in a *Pinus chiapensis* tree that was adjacent to the tree of the same species where she was originally found. Albeit limited, these data suggest that *A. cunemica* sp. nov. can be relatively sedentary at least some months of the year.

Based on our field sampling, arboreal or semi-arboreal herpetofauna with which *A. cunemica* sp. nov. occurs in sympatry (within 2 airline km) include *Sceloporus internasalis* (MZFZ 4477), *Imantodes gemmistratus* (MZFZ 4562), *Bolitoglossa rufescens* (MZFZ 4473), *Bromeliohyla bromeliacia* (MZ-UNICACH 68–79) [93], and *Smilisca baudinii* (MZFZ 4561, 4565). One or more of these species might be a food source for the new species, and *I. gemmistratus* could conceivably eat young *A. cunemica* sp. nov. Including previous records [88], we also documented sympatric co-occurrence of the following reptiles: *Anolis tropidonotus* (= *A. spilorhipis* or *A. t. spilorhipis* of some authors; MZFZ 4559–4560), *Scincella incerta* (MZFZ 4472), *Drymobius margaritiferus* (MZ-UNICACH 194) [88], *Geophis carinosus* (MZFZ 4475–4476), *Ninia diademata* (MZ-UNICACH 196, 376), *Pituophis lineaticollis* (MZ-UNICACH 198) [88], *Rhadinaea* [= *Rhadinella*] *godmani* (MZFC-HE 29997) [94], *Tantilla schistosa* MZFZ 4471, *Thamnophis cyrtopsis* (MZ-UNICACH 378) [88], *Micrurus elegans* (MZFZ 4474); and the following amphibians: *Incilius valliceps* (MZ-UNICACH 95–98) [88], *Craugastor lineatus* (MZ-UNICACH 454) [88], *C. pygmaeus* (MZFZ 4563–4564), and *Rana brownorum* (MZ-UNICACH 155–157) [88]. Further sampling will likely show *A. cunemica* sp. nov. to co-occur with the terrestrial congener *Abronia temporalis*, which has been collected less than 8 km to the northeast (MVZ 272336).

## Conservation

*Abronia cunemica* sp. nov. is unrecorded from any protected areas. However, several forest parcels within a 5 km radius of the type locality have been designated as small reserves by residents of Coapilla. These patches could support populations of the species. Upon touring one such parcel, we found a more intact forest canopy compared to other lands that we visited, but the understory was highly modified by widespread cutting of saplings (probably for firewood). The Parque Ecoturístico Laguna Verde is another area managed by the Ejido of Coapilla where *A. cunemica* sp. nov likely occurs, and where relatively environmentally friendly ecotourism activities are promoted by the community. However, socioeconomic problems that are well-documented in other protected areas of Chiapas also affect this ecotourism park and its surroundings [95–98]. Lastly, an additional 101-ha protected area called the Zona Sujeta a Conservación Ecológica Tzama Cum Pümy, which lies some 7 km from the type locality of *A. cunemica* sp. nov., could also support the species [99–101]. Future surveys, in close partnership with community leaders in Coapilla and Tapalapa, are necessary to confirm these suspicions.

The forest on the tableland where *A. cunemica* sp. nov. lives is generally perturbed to some degree (e.g., Fig 9A). Much of the landscape has been converted to a savannah, with widely-spaced trees and a grassy understory with few shrubs and saplings. On multiple occasions, we

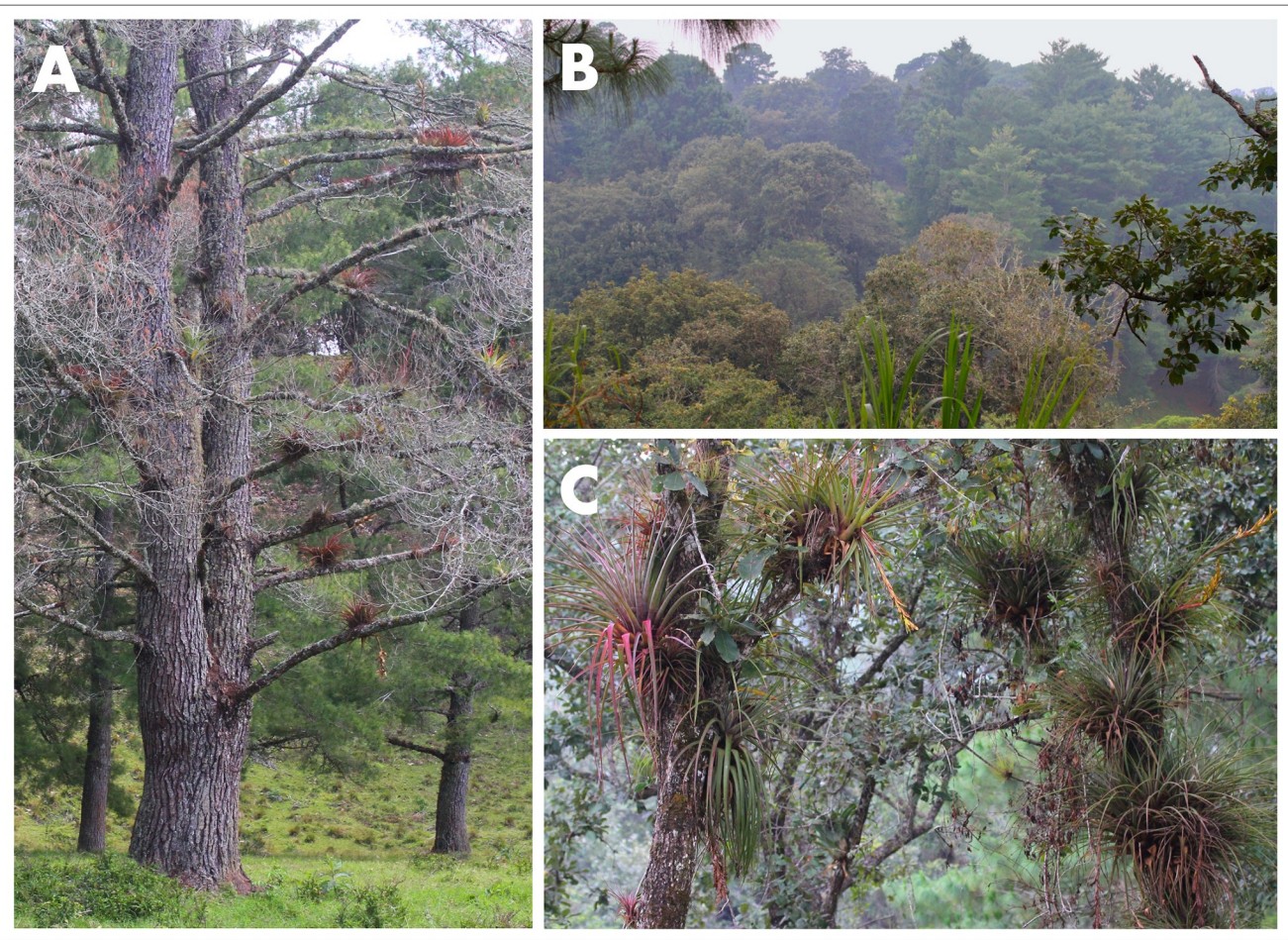

**Fig 9. Habitat of *Abronia cunemica* sp. nov. near Coapilla, Chiapas, Mexico.** (A) Live and dead *Pinus chiapensis* trees in a cattle pasture, (B) intact *Pinus-Quercus* forest, (C) microhabitat of *Quercus* spp. trees laden with epiphytes. Photographs taken by Adam G. Clause on 19 February 2022 (A) and 14 August 2021 (B–C).

also observed trucks loaded with large freshly-cut pine logs leaving the area. Within 100 m of all sites where we found *A. cunemica* sp. nov., we saw agricultural, livestock, and forestry activities taking place. The absence of intact forest and ongoing loss of trees are nonetheless tempered by three factors. First, many communally-owned parcels in the Ejido of Coapilla are under active forest management. Based on public signage that we observed, this management approach appears to disallow clear-cuts in favor of selective logging on a cyclical basis across a mosaic of designated parcels. Second, our surveys revealed that adult female and juvenile *A. cunemica* sp. nov. can live in remnant trees growing within pastureland. The species thus might have substantial tolerance to forest disturbance, similar to reports for a few other arboreal species of *Abronia* [102]. Third, individuals of *A. cunemica* sp. nov. appear to spend most of their lives hidden in the forest canopy, where they are inherently sheltered from direct human influence. For these reasons, despite ongoing forest loss that clearly eliminates habitat for this species and imperils its survival, we remain cautiously hopeful about its long-term prognosis.

There are additional reasons for guarded optimism about the future for *A. cunemica* sp. nov., in our judgement. Because the species lives in mid-elevation portions of the Northern

Highlands, it probably has room to expand upward in elevation as the pine-oak forest is pushed upward by climate change [28, 30, but see 31]. This is unlike its close relative *A. morenica*, which appears to only live in high-elevation mountaintop cloud forest and is thus more strongly threatened by habitat loss driven by climate change [44]. Furthermore, although we encountered a few people who mistakenly feared *A. cunemica* sp. nov., believing it to be venomous, the overall lack of familiarity with the species by community members suggests that populations are not subject to serious pressure from unwarranted killing. Lastly, although arboreal members of the genus *Abronia* are increasingly targeted for the international illegal pet trade [103–105], we suspect that the extraordinary difficulty which we experienced in finding this species will reduce the intensity of this threat. Nonetheless, to further reduce any possible poaching pressure, we have intentionally masked the localities mentioned in this paper by (1) rounding GPS coordinates to the nearest hundredth of a degree, and (2) not reporting distance or direction from the nearest town [106].

Although all conclusions regarding threats are based on limited data and thus represent informed conjecture, we consider it justifiable to provisionally categorize *A. cunemica* sp. nov. on three lists of imperiled species. For the Red List of Threatened Species of the International Union for Conservation of Nature, we recommend that *A. cunemica* sp. nov be assessed as Endangered (B1ab[iii,v]+2ab[iii,v]) [107]. In meeting these criteria for Endangered status, we consider the species to have an extent of occurrence and area of occupancy of much less than 5,000 km$^2$ and 500 km$^2$, respectively; to be known from fewer than five locations; and to be experiencing a inferred decline in the extent of habitat, quality of habitat, and number of mature individuals, due to cutting of trees and possible loss of individuals to targeted killing and poaching [107]. Similarly, we calculate that *A. cunemica* sp. nov. has an Environmental Vulnerability Score (EVS, see Wilson et al. [108]) of 18 out of 20, which places the species in the High Vulnerability category. We derive this total score from assessed criteria on geographic distribution (6 points, for having a distribution limited to Mexico in the vicinity of the type locality), ecological distribution (8 points, for being currently known from only one vegetation formation), and degree of human persecution (4 points, for having arboreal habits yet thought to be harmful to people, and might be killed if encountered). Finally, we propose that *A. cunemica* sp. nov. be categorized as Amenazada (Threatened) on the Norma Oficial Mexicana list [109], due to the species meeting Criteria A.(I)+B.(II)+C.(II)+D.(III) as defined by the Anexo Normativo I or Método de Evaluación del Riesgo de Extinción de las Especies Silvestres en México (MER) [110]. This is the only list of at-risk species that is legally recognized by the Mexican government. We suggest this categorization based on *A. cunemica* sp. nov. having the following characteristics: a very restricted distribution (4 points) that encompasses less than 5% of the national territory of Mexico; an intermediate or limiting habitat (2 points) with respect to the requirements for the natural development of the species, due to its presumed restriction to mid-elevation forests; a medium vulnerability (2 points) intrinsic to the biology of the species, due to its specialized natural history that depends on trees but allows the species to persist in fragmented forests; and a medium impact (3 points) by humans on the species due to pressure from human settlements, moderate habitat fragmentation, and the potential use, trade and trafficking of the species. The total score of 11 points justifies an assignment to the category of Amenazada [109].

## Discussion

Newly discovered species of vertebrates tend to have small geographical distributions, which generally place those species at imminent risk of extinction because range size strongly influences endangerment [111, 112]. These patterns are particularly evident in lizards, for which

the rate of description of new species is rapidly accelerating [113]. Our discovery of *A. cunemica* sp. nov., a microendemic lizard that is endangered, is consistent with these global trends.

With the recognition of *A. cunemica* sp. nov., the number of described species in the genus is increased to 42, of which 32 are considered arboreal [45]. It is the 28th species of *Abronia* known from Mexico [45, 114]. The total number of described native lizards (excluding Serpentes) in Chiapas is now 94 [44, 51, 115–119]. Of these lizard species, 10 are endemic to the state of Chiapas [44, 52].

The phylogenetic position of the clade composed of *A. cunemica* sp. nov., *A. morenica*, and *A. ornelasi* within the Eastern clade of *Abronia* (as defined by Gutiérrez-Rodríguez et al. [38]) is congruent with their geographic distributions east of the Isthmus of Tehuantepec. The topology of our phylogenetic tree opens three possible higher-level taxonomic arrangements for the *A. cunemica* sp. nov./*A. morenica*/*A. ornelasi* clade. First, the clade could be subsumed within Group VII as previously defined [38]. Second, the clade could be recognized as a new, ninth group of *Abronia* (Group IX). Third, the clade could be recognized as two new groups of *Abronia*: Group IX (that would include *A. ornelasi* and tentatively *A. reidi*, which together correspond to the subgenus *Abaculabronia* [39]), and Group X (that would include *A. cunemica* sp. nov., *A. morenica*, and tentatively *A. frosti*, *A. montecristoi*, and *A. salvadorensis*, which together correspond to the subgenus *Lissabronia* [1, 44]). Because no genetic samples of *A. reidi* (type species of *Abaculabronia*) nor *A. salvadorensis* (type species of *Lissabronia*) were available for inclusion within our molecular phylogeny, we lack confidence in choosing among these three taxonomic options. However, we provisionally prefer the third option, and thus assign *A. cunemica* sp. nov. to the subgenus *Lissabronia*. We await the availability of a molecular phylogeny with more comprehensive sampling of aboreal *Abronia* species to test this assignment.

With the provisional addition of *A. cunemica* sp. nov. the content of *Lissabronia* is increased to five species, with the others being *A. frosti* [1] from Guatemala, *A. montecristoi* [120] from El Salvador and Honduras; *A. morenica* [44] from Mexico, and *A. salvadorensis* [120] from Honduras (type species). Within *Lissabronia*, *A. cunemica* sp. nov. appears to be most morphologically similar to *A. morenica* and *A. salvadorensis* among all recognized *Abronia* (Table 1). This close resemblance to *A. salvadorensis* is surprising because it is separated by over 600 km from *A. cunemica* sp. nov., making them the most geographically distant of all *Lissabronia*. The new species can be readily distinguished from *A. morenica*, which occurs 110 km to the south and is the geographically closest species of *Lissabronia*, as follows (*A. morenica* character state in parentheses): 35–39 transverse dorsal scale rows (vs. 30–35), dorsum of head pale yellow with distinct dark markings (vs. never yellow and dark markings absent or faint), flanks sometimes mostly pale yellow, but yellow or orange posterior blotch on flank scales absent (vs. present), and adult snout-to-vent length 107–127 mm (vs. 92–93 mm). These physical differences support our conclusion that the two known populations of *A. cunemica* sp. nov. and *A. morenica* are not members of the same species. Cumulatively, our phylogenetic, population genetic, and species delimitation analyses corroborate this proposed recognition of *A. cunemica* sp. nov. as a species distinct from *A. morenica*. We recognize, however, that these two taxa are not deeply divergent genetically. We also acknowledge an alternative interpretation of the genetic data: that *A. cunemica* sp. nov. simply represents the second known population of *A. morenica*. Given their dramatic biogeographic isolation, with the inhospitable semi-arid Central Depression almost certainly preventing ongoing gene flow (Fig 8), in our view these two populations are on separate evolutionary trajectories. Hence, we reject this alternative interpretation and instead consider the Coapilla population to be a full species.

We take this opportunity to revisit a few troublesome character states for the paratypes of *A. morenica*, which were inadvertently mis-scored in the original description of paratype

variation [44]. These errors do not affect the diagnosis of *A. morenica*, nor do they affect comparisons with *A. cunemica* sp. nov. Nonetheless, we correct them in this study for transparency and to ensure consistent morphological interpretation for these seemingly closely related taxa. The updated character state scores are as follows: secondary temporal scales 3/3, with the first and second scales on the left side (MZFC-HE 33488) or both sides (MZFC-HE 33484) separated by an anteriorly displaced second tertiary temporal; tertiary temporals 4–5 ($\bar{X} = 4.6$); supralabials 9–11 ($\bar{X} = 9.8$), with the antepenultimate being the posteriormost to reach the orbit in MZFC-HE 33485, 33488, and 34400; infralabials 7–10 ($\bar{X} = 8.3$); sublabials 4–6 ($\bar{X} = 4.6$).

Unlike the majority of lizard and snake species recently proposed for recognition in Mexico, but similar to other newly announced members of its genus [44, 45], *A. cunemica* sp. nov. was previously completely unknown to science. The discovery of *A. cunemica* sp. nov. in the Northern Highlands of Chiapas was particularly unexpected because of the short distance separating Coapilla from known populations of *A. lythrochila* in the same physiographic region. The latter species, which is a member of the *Auriculabronia* subgenus and is thus not a close relative of *A. cunemica* sp. nov., occurs less than 40 km to the east-southeast near the town of Jitotol [60–62] (Fig 8). The intervening slopes and ridges of the Tapalapa–Rayón–Pueblo Nuevo Solistahuacán corridor support extensive tracts of mesic forest that appear suitable for arboreal members of the genus *Abronia*. New surveys of this poorly explored corridor will undoubtedly further reduce the distance separating these two species. Nonetheless, low-elevation valleys in the vicinity of Rayón converge to create a corridor of 1500 m elevation that cleanly divides the higher mountains to the east and west. We consider it plausible that this division represents a physical barrier that prevents contact between *A. cunemica* sp. nov. and *A. lythrochila*, but see García-Padilla and Escalante-Pliego [121].

Despite being accessible by a paved road, which makes Coapilla less than a 3-hr drive from the largest city in Chiapas (Tuxtla Gutiérrez), the vertebrate biodiversity in the Coapilla area remains incompletely understood. Prior to the work published herein, only two studies of the local herpetofauna were available [88, 93]. The local mammal assemblage is also poorly researched. A total of 71 mammal species and subspecies have been documented in the Municipality of Coapilla [122, 123]. Notably, this includes the first and second records from Chiapas for the bat species *Nyctinomops macrotis* and *N. laticaudatus*, respectively [122]. Remarkably, we are aware of no published studies of birds in the area. Informal reports nonetheless indicate that at least two imperiled and striking birds (*Pharomachrus mocinno* and *Setophaga chrysoparia*) migrate seasonally to live in or near Coapilla [124].

Building on the presence of these rarely documented bats and at-risk birds, we advocate that *A. cunemica* sp. nov. be promoted as a conservation flagship for the region. As a microendemic member of a group of striking, iconic lizards, we consider its potential as a flagship to be strong. Nonetheless, we also acknowledge the importance of educational activities in the region to help prevent illegal trafficking of the species.

The need for greater conservation efforts in the Northern Highlands is clear. Although Chiapas supports the greatest extent of cloud forest of any state in Mexico (6,037 km$^2$), with around a quarter of those forests currently protected [28], most of those Protected Natural Areas are found in the Sierra Madre de Chiapas. In fact, only one internationally recognized protected area includes lands above 1500 m elevation in the Northern Highlands: the 101-hectare Zona Sujeta a Conservación Ecológica Tzama Cum Pümy [99–101]. The mid- to high-elevation parts of this physiographic region, which has experienced substantial deforestation [125], are thus a big gap in the existing network of protected areas in Chiapas [126]. Creating new reserves, and strengthening societal investment in areas that already receive some degree

of local-level protected status, is an urgent need [125, 127]. Partnerships to build community buy-in among local stakeholders must be a part of any such programs [125, 128, 129]. Given that land use change is the leading driver of biodiversity loss globally [130] we encourage grass-roots efforts to safeguard the forests where *A. cunemica* sp. nov. lives. Such efforts would bring long-overdue conservation attention to the remarkable biodiversity of the Northern Highlands of Chiapas.

## Supporting information

**S1 Appendix. Complete Spanish translation of article.**
(PDF)

**S2 Appendix. Locality information and GenBank accession numbers for samples used in this study.**
(XLSX)

## Acknowledgments

We thank José Manuel Aranda-Coello for generously sharing relevant information. For help with fieldwork, we are indebted to our team members Emmanuel Javier-Vázquez, Candelario Cundapí-Pérez, Marcos Joaquín Fitz-Pérez, Ana Reyna Pale Morales, José Manuel Toledo-Morales, Víctor Vásquez-Cruz, Maisie G. MacKnight, Jorge Arturo Hidalgo-García, Daniel Lara-Tufiño, and Justin K. Clause. Uri Omar García-Vázquez provided important logistical support, for which we are grateful. Our thanks to Francisco Hernández-Najarro and Oscar Farrera-Sarmiento for plant identifications, to Ernesto Velázquez-Velázquez for authorizing our access to the MZ-UNICACH collection, to Martha Erika Hernández de la Cruz and Gloria Pérez for information about the name of Coapilla in Zoque, and to Marcy Kinsey for help with color interpretation. All analyses of the molecular data were performed at the Mana high performing computing cluster of the University of Hawaii. We thank Jorge Gutiérrez-Rodríguez for generating ddRADseq data for some of the specimens used in our analyses.

## Author Contributions

**Conceptualization:** Adam G. Clause, Roberto Luna-Reyes, Israel Solano-Zavaleta.

**Data curation:** Adam G. Clause, Oscar M. Mendoza-Velázquez, Adrián Nieto-Montes de Oca.

**Formal analysis:** Adrián Nieto-Montes de Oca.

**Funding acquisition:** Adam G. Clause, Adrián Nieto-Montes de Oca.

**Investigation:** Adam G. Clause, Roberto Luna-Reyes, Oscar M. Mendoza-Velázquez, Adrián Nieto-Montes de Oca, Israel Solano-Zavaleta.

**Methodology:** Adam G. Clause, Adrián Nieto-Montes de Oca.

**Supervision:** Adam G. Clause.

**Visualization:** Adam G. Clause, Adrián Nieto-Montes de Oca, Israel Solano-Zavaleta.

**Writing – original draft:** Adam G. Clause, Roberto Luna-Reyes, Oscar M. Mendoza-Velázquez, Adrián Nieto-Montes de Oca, Israel Solano-Zavaleta.

**Writing – review & editing:** Adam G. Clause, Roberto Luna-Reyes, Oscar M. Mendoza-Velázquez, Adrián Nieto-Montes de Oca, Israel Solano-Zavaleta.

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
