## [Decision Letter · Decision Letter 0]

16 Mar 2023

PONE-D-22-34275Bridging the gap: A new species of arboreal Abronia (Squamata: Anguidae) from the Northern Highlands of Chiapas, MexicoPLOS ONE

Dear Dr. Clause,

Thank you for submitting your manuscript to PLOS ONE. After careful consideration, we feel that it has merit but does not fully meet PLOS ONE’s publication criteria as it currently stands. Therefore, we invite you to submit a revised version of the manuscript that addresses the points raised during the review process.

We look forward to receiving your revised manuscript.

Kind regards,

Tzen-Yuh Chiang

Academic Editor

PLOS ONE

Journal Requirements:

2. We note that you have referenced (unpublished on page 25) which has currently not yet been accepted for publication. Please respond by return e-mail with a copy of your updated manuscript to include to remove this from your References and amend this to state in the body of your manuscript: (ie “Bewick et al. [Unpublished]”) as detailed online in our guide for authors

http://journals.plos.org/plosone/s/submission-guidelines#loc-reference-style.   We can then upload this to your submission on your behalf.

3. We note that Figure 4 in your submission contain [map/satellite] images which may be copyrighted. All PLOS content is published under the Creative Commons Attribution License (CC BY 4.0), which means that the manuscript, images, and Supporting Information files will be freely available online, and any third party is permitted to access, download, copy, distribute, and use these materials in any way, even commercially, with proper attribution. For these reasons, we cannot publish previously copyrighted maps or satellite images created using proprietary data, such as Google software (Google Maps, Street View, and Earth). For more information, see our copyright guidelines: http://journals.plos.org/plosone/s/licenses-and-copyright.

a. You may seek permission from the original copyright holder of Figure 4  to publish the content specifically under the CC BY 4.0 license.  

Reviewers' comments:

Reviewer's Responses to Questions

**Comments to the Author**

1. Is the manuscript technically sound, and do the data support the conclusions?

Reviewer #1: Yes

Reviewer #2: Yes

Reviewer #3: Partly

Reviewer #4: Yes

Reviewer #5: Yes

Reviewer #6: Partly

2. Has the statistical analysis been performed appropriately and rigorously? 

Reviewer #1: Yes

Reviewer #2: N/A

Reviewer #3: N/A

Reviewer #4: N/A

Reviewer #5: Yes

Reviewer #6: N/A

3. Have the authors made all data underlying the findings in their manuscript fully available?

Reviewer #1: Yes

Reviewer #2: No

Reviewer #3: Yes

Reviewer #4: Yes

Reviewer #5: Yes

Reviewer #6: Yes

4. Is the manuscript presented in an intelligible fashion and written in standard English?

Reviewer #1: Yes

Reviewer #2: Yes

Reviewer #3: Yes

Reviewer #4: Yes

Reviewer #5: Yes

Reviewer #6: Yes

5. Review Comments to the Author

Reviewer #1: This manuscript is a well-documented description of an interesting new species of lizard from the northern highlands of Chiapas. As such, it is an important contribution to the herpetology of the area. It appears to me to meet all publication criteria laid out by the journal.

I note only one thing tht might need attention. There is a bit of taxonomic problem that surfaces on p. 5 (line 97) where it is stated that Abronia lythrochila is a second species of Abronia occurring in the northern highlands of Chiapas. I am of the opinion that A. lythrochila is a junior synonym of A. ochoterenae. This arrangement has not been published and I discuss it in an unpublished book manuscript that will not be out for a while. I am uncertain if the authors of this Abroniadescription would agree with me on this or not, but if not, it begs the question—where do they allocate A. ochoterenai? Given that some of these data are unpublished, I don't know how (or if) the authors want to handle it. Certainly, if they arrive at the same conclusion about A. lythrochila and would like to publish this, I would have no qualms. One way or the other, I think A. ochoterenai occurs in the northern highlands and should be delt with.

I have practically nothing to add to this remarkably well-written paper that was obviously thoroughly scrutinized prior to submission. I enthusiastically recommend it for publication.

Reviewer #2: This manuscript describes a new species of Abronia from Chiapas. It is well written and well done. The finding of a new species of Abronia highlights the likely hidden diversity found in Mexico, especially in the topographically heterogenous regions. I have only a few suggestions to improve the manuscript.

Line 110: There is actually very little about how specimens were collected. It would be useful to provide more information about how you searched for and caught the specimens. Some of this information is provided scattered throughout the rest of the paper, but it would be good to provide details here.

Lines 134-136: Were measurements made on living or preserved specimens?

Line 271: I would change the “the four currently” to “all four currently” to clarify.

Line 334: How did you define colors?

Lines 378 and throughout: Please provide a measure of variation (e.g., SE or SD).

Lines 555-557: This statement seems a bit premature. Given the amount of time between observations you don’t know that it didn’t move further since it is based on 2 observations.

Line 601: I would change “thus has substantial tolerance” to “thus may have substantial tolerance”. You are drawing too firm of a conclusion based on very few observations and little knowledge. I would be more circumspect in your conclusions.

Line 629: This seems to contradict the previous paragraph where you said killing and poaching was unlikely.

Reviewer #3: The manuscript is nice, clearly written, based on years of fieldwork, presents interesting finding of rare animal population. Unfortunately, it completely lacking and grounds for description of newly discovered isolated population as a new species. No statistical analysis to assess morphological difference from sister species is provided and detailed description of individuals do not illustrate magnitude of these differences. Sadly, no genetic data is provided. Even in the absence of simple barcode sequences of presumably related species, providing sequence of a new species would simplify further work towards taxonomic integrity in the attractive and taxon rich group. I believe that this all make the quality of research too low to the current informal standards of new taxa description and jeopardize integrity of hypothesis of new species.

Reviewer #4: Comments

Dear Editor,

I have read the MS entitled “Bridging the gap: A new species of arboreal Abronia (Squamata: Anguidae) from the Northern Highlands of Chiapas, Mexico” by Adam G. Clause, Roberto Luna-Reyes,¶ Oscar M. Mendoza-Velázquez¶, Israel Solano-

Zavaleta.

This a good paper, description of a new species of Abronia; it is well written, good material, and the species is described based in morphological characteristics (diagnostic characteristics), from my point of view, the use of morphological methods for describing new species, it is good as long as the sample size indicates the diagnostic characters that indicate that they represent a new species and this is shown here; otherwise the authors would have to use molecular methods (molecular biology). Therefore, my comments on this work are very soft, indicating some suggestions that the authors can attend to if they wish.

Title- why Bridging the gap? Could you explain it a little bit in introduction?

Introduction. Could you explain a little more about the Mesoamerican region in terms of it being a center of diversification and its importance from a biogeographical point of view, its endemicity. The importance of cloud forest and pine-oak forest environments, which are the environments in which Abronia species inhabit. It is important to give it a sense of conservation and protection to maintain biodiversity (see comments in text).

Methods- Holotype- The morphological part that comes from the holotype is very heavy, perhaps mentioning the diagnostic characters that indicate being a new species will be enough and comparing these with the characters of another or other species closer to the genus Abronia would be enough, and in a section (appendix) you can put the other characters that are important to recognize the new species, but you could solve in this way putting a large number of characters that the reader is lost (it is similar en results section).

Results- see comments on text.

Discussion- See comments on text. It is important to point out the care of the environment in these areas where the diversification of the species occurs, focusing on the conservation of biodiversity, specifically in the genus Abronia, whose species are arboreal; in these environments, the anthropic effect is ending the vegetation cover, eliminating the refuge(s) of the species.

Reviewer #5: I this study, the authors describe a new species of the arboreal lizard genus Abronia. Most species in this genus are endangered, including this new species. The text is well structured, and all the relevant characters for the description of the species were considered. Also, important information about the geogrpahic distribution of the Abronia species from the state of Chiapas, Mexico, is provided. I believe this work deserves to be considered for its publication in PLoS ONE as it is.

Reviewer #6: This manuscript describes a new species of Arboreal Alligator lizard (Abronia) from the Northern highlands of Mexican Chiapas. The authors identify diagnostic combinations of scalation characters and geographic isolation as evidence for these animals belonging to a new species. The paper is well written with good figures,  the description and diagnoses are well laid out and appropriate and, as Abronia species have famously restricted ranges, this almost certainly will prove to be a new species. However, if this is published as is, it leaves some shortcomings that will need to addressed up by future studies, and I worry that the authors have not done enough to warrant this being published in PlosOne. 

The authors employ De Quiroz’s evolutionary species concept and identify 1) fixed morphological differences between the new species and select congeners, and 2) geographic isolation as their delimitation methods. No molecular phylogenetics were attempted, despite tissue being available from the new species and the putative sister species (A.morenica), and mtDNA sequences are available on genbank from the geographically proximal A lythrochila, and the lissabronine A. frosti, as well as several other species (eg A. moreletii, A. graminea, A. monticola, A. cuchumatanus, A. gadovii, A. aurita, A. lythrochila, A. campbelli, A. fimbriata, A. sp., A. matudai, A. anzuetoi, A. mixteca, A. ornelasi, Gerrhonotus liocephalus, Barisia imbricata, A. chiszari, and A. oaxacae), not to mention the data from an unpublished  Masters thesis from the anchor author, and a well sampled phylogenomic analysis that was also coauthored by the anchor author. I understand that that there are challenges to incorporating novel sequences into existing datasets, especially genomic ones, but if the authors were  able to incorporate some genetic data it would really strengthen the manuscript and further untangle the Abronia relationships.

The Authors assign the new species to a subgenus without any phylogenetic analysis to confirm this statement. Furthermore, the proposed assignment of the new species is based on a taxonomy that is has been shown, through molecular phylogenetics, to not consistently represent the evolutionary history of the group. The most recent phylogeny by Guitérrez-Rodríguez et al. 2020 suggests that many /most of the subgenera proposed by Campbell and Frost (1993) were paraphyletic, and even showed that Abronia as a whole was paraphyletic! Although the authors follow the taxonomic assessment of the 2020 paper and refer to the 11 clades, they still use the groupings of Campbell and Frost to describe the phylogenetic placement of the new species. They place it the subgenus Lissabronia, and point to its geographic isolation from all other members of the subgenus as evidence of its evolutionary isolation, dismissing the possibility that the new species could be associated with the geographically proximal A.lythrochila. Perhaps this new species is unambiguously attributable to Lissabronia (though the subgenus does lack autapomorphies-not synapomorphies as stated), but a phylogenetic assessment, even morphological, would help make that point. Furthermore,  I expect that some ecological niche models would help confirm that the species is geographically isolated from the nearby A.lythrochila, especially using data layers that reveal vegetation cover, which seems to be an important factor in Abronia distribution.

In summary, I am sure that this species assignment will prove valid, but I am concerned that the reliance of scalation characters to delimit the species and assign to a subgenus is undermined by the inconsistencies the morphological and molecular phylogenies. The authors do have tissues for this and the putative sister species, and I would strongly recommend that they incorporate some kind of phylogenetic analysis into this description.

6. PLOS authors have the option to publish the peer review history of their article (what does this mean?). If published, this will include your full peer review and any attached files.

Reviewer #1: **Yes: **Jonathan A. Campbell

Reviewer #2: No

Reviewer #3: No

Reviewer #4: No

Reviewer #5: No

Reviewer #6: No

---

## [Author Response · Author response to Decision Letter 0]

16 Sep 2023

(Reviewer #1, Jonathan A. Campbell): This manuscript is a well-documented description of an interesting new species of lizard from the northern highlands of Chiapas. As such, it is an important contribution to the herpetology of the area. It appears to me to meet all publication criteria laid out by the journal.

Author Response: We are grateful for the reviewer’s positive assessment of our work. 

(Reviewer #1): I note only one thing that might need attention. There is a bit of taxonomic problem that surfaces on p. 5 (line 97) where it is stated that Abronia lythrochila is a second species of Abronia occurring in the northern highlands of Chiapas. I am of the opinion that A. lythrochila is a junior synonym of A. ochoterenae. This arrangement has not been published and I discuss it in an unpublished book manuscript that will not be out for a while. I am uncertain if the authors of this Abronia description would agree with me on this or not, but if not, it begs the question—where do they allocate A. ochoterenai? Given that some of these data are unpublished, I don't know how (or if) the authors want to handle it. Certainly, if they arrive at the same conclusion about A. lythrochila and would like to publish this, I would have no qualms. One way or the other, I think A. ochoterenai occurs in the northern highlands and should be delt with.

Author Response: We appreciate the reviewer’s comment and concerns regarding Abronia ochoterenai. Although the type and only confirmed locality for A. ochoterenai (as re-defined by Campbell & Frost, 1993) is vague (see Peterson and Nieto-Montes de Oca, 1996), the locality lies in the southern Central Plateau or possibly the southwestern Eastern Highlands physiographic region of Chiapas as defined by Breedlove (1981) and Johnson et al. (2015). As such, A. ochoterenai is over 100 km from the new species we propose from Coapilla, which is known only from the Northern Highlands physiographic region of Chiapas. We certainly agree with the reviewer that A. lythrochila could be a junior synonym of A. ochoterenai. This conclusion was also suggested by Gutiérrez-Rodríguez et al. (2021), based on their recent molecular analysis that included a single sample from each of those two taxa. However, conclusively supporting a taxonomic proposal to synonymize A. lythrochila with A. ochoterenai will require rigorous morphological and molecular analyses of most of the available material for both taxa, and a detailed re-description of A. ochoterenai. This would easily constitute a separate paper. Furthermore, for the purposes of our current manuscript, in which our core goal is to demonstrate the distinctiveness of the proposed new species of Abronia from Coapilla, we have added a molecular phylogeny that includes a sample of presumed A. ochoterenai from near the type locality—and it is not closely related to the Abronia from Coapilla. With the inclusion of this sample, we consider that our revised manuscript addresses the reviewer’s concerns about A. ochoterenai.

(Reviewer #1): I have practically nothing to add to this remarkably well-written paper that was obviously thoroughly scrutinized prior to submission. I enthusiastically recommend it for publication.

Author Response: Once again, we are grateful for the reviewer’s feedback and kind words.

(Reviewer #2, anonymous): This manuscript describes a new species of Abronia from Chiapas. It is well written and well done. The finding of a new species of Abronia highlights the likely hidden diversity found in Mexico, especially in the topographically heterogenous regions. I have only a few suggestions to improve the manuscript.

Author Response: We thank the reviewer for their generous assessment and helpful suggestions.

(Reviewer #2) Line 110: There is actually very little about how specimens were collected. It would be useful to provide more information about how you searched for and caught the specimens. Some of this information is provided scattered throughout the rest of the paper, but it would be good to provide details here.

Author Response: Thank you, we added an opening sentence to this section that explains how we searched for and collected the specimens. Additional information about our search effort is consolidated in the third paragraph of the Distribution and Ecology section of the manuscript.

(Reviewer #2) Lines 134-136: Were measurements made on living or preserved specimens?

Author Response: We took measurements after euthanasia but prior to formalin fixation, and we now specify that in the Morphology section of the manuscript.

(Reviewer #2) Line 271: I would change the “the four currently” to “all four currently” to clarify.

Author Response: Done, thank you.

(Reviewer #2) Line 334: How did you define colors?

Author Response: We did not use a highly specific color guide to define colors, and instead restricted our color terminology to common-knowledge names and phrases (“pale yellow,” “dark brown,” etc.) due to the lack of diagnostic importance of subtle color differences within the genus Abronia. 

(Reviewer #2) Lines 378 and throughout: Please provide a measure of variation (e.g., SE or SD).

Author Response: Because of the small sample sizes (n ≤ 4), and because of the minimal variation in each feature (the upper and lower bounds of most measurements differ by just 1 mm or 1 scale), we argue that calculating SE or SD provides no relevant information. Furthermore, only range and mean values are given in previously published descriptions of new alligator lizard species with similarly small sample sizes and similarly minimal variation. We have chosen to follow that convention in our manuscript. However, if the editor or reviewer insists otherwise, we will consider changing our position on this matter. Thank you.

(Reviewer #2) Lines 555-557: This statement seems a bit premature. Given the amount of time between observations you don’t know that it didn’t move further since it is based on 2 observations.

Author Response: Agreed, change made.

(Reviewer #2) Line 601: I would change “thus has substantial tolerance” to “thus may have substantial tolerance”. You are drawing too firm of a conclusion based on very few observations and little knowledge. I would be more circumspect in your conclusions.

Author Response: Agreed, change made.

(Reviewer #2) Line 629: This seems to contradict the previous paragraph where you said killing and poaching was unlikely.

Author Response: Agreed, we have now modified the sentence to indicate that loss of individuals due to targeted killing and poaching is “possible.” 

(Reviewer #3, anonymous): The manuscript is nice, clearly written, based on years of fieldwork, presents interesting finding of rare animal population. Unfortunately, it completely lacking and grounds for description of newly discovered isolated population as a new species. No statistical analysis to assess morphological difference from sister species is provided and detailed description of individuals do not illustrate magnitude of these differences. Sadly, no genetic data is provided. Even in the absence of simple barcode sequences of presumably related species, providing sequence of a new species would simplify further work towards taxonomic integrity in the attractive and taxon rich group. I believe that this all make the quality of research too low to the current informal standards of new taxa description and jeopardize integrity of hypothesis of new species.

Author Response: We appreciate the reviewer’s concerns about the lack of genetic data in our manuscript, and we agree that this was a shortcoming of our initial submission. We have now added a genomic dataset, which we hope will fully resolve these concerns. Regarding the need for statistical analyses to assess the morphological differences of the proposed new species relative to its putative sister species, we argue that such analyses would be uninformative. As we show in the Diagnosis section of our manuscript, the morphological features that differentiate the two taxa are non-overlapping meristic characters (scale counts, size, and color pattern), that are highly consistent and vary minimally. Hence, we are unaware of any statistical analyses that would shed further light on the diagnostic value of those characters. Nonetheless, if there are specific analyses that the reviewer or editor wishes to propose for our morphological data, we will certainly consider that request.

(Reviewer #4, anonymous): This a good paper, description of a new species of Abronia; it is well written, good material, and the species is described based in morphological characteristics (diagnostic characteristics), from my point of view, the use of morphological methods for describing new species, it is good as long as the sample size indicates the diagnostic characters that indicate that they represent a new species and this is shown here; otherwise the authors would have to use molecular methods (molecular biology). Therefore, my comments on this work are very soft, indicating some suggestions that the authors can attend to if they wish.

Author Response: Thank you, we appreciate this useful feedback on our manuscript.

(Reviewer #4): Title- why Bridging the gap? Could you explain it a little bit in introduction?

Author Response: Thank you, we have now changed our language in the Introduction from “blank spot” to “gap” to better echo the title.

(Reviewer #4): Introduction. Could you explain a little more about the Mesoamerican region in terms of it being a center of diversification and its importance from a biogeographical point of view, its endemicity. The importance of cloud forest and pine-oak forest environments, which are the environments in which Abronia species inhabit. It is important to give it a sense of conservation and protection to maintain biodiversity (see comments in text).

Author Response: Thank you, we explain in depth the biogeographical importance of and diversification within the Mesoamerican region in the first paragraph of our Introduction. In that same paragraph, we provide 35 citations to published works to which readers can refer for more information.

(Reviewer #4) Lines 55–57: Could you please add some examples of species that have arisen in these areas, which represent centers of speciation?

Author Response: In the citations at the end of this sentence, we offer 7 published papers to which readers can refer if they are interested in particular species or species groups that have diversified in this area.

(Reviewer #4) Lines 71–75: Could you add a paragraph about some ecological aspects of the group, for example, all species are arboreal, their distribution is just cloud forest or is wider.

Author Response: We provide these details about the ecology of Abronia in two detailed sentences, which include 4 citations to published works, on lines 78–81 of our revised manuscript.

(Reviewer #4): Methods- Holotype- The morphological part that comes from the holotype is very heavy, perhaps mentioning the diagnostic characters that indicate being a new species will be enough and comparing these with the characters of another or other species closer to the genus Abronia would be enough, and in a section (appendix) you can put the other characters that are important to recognize the new species, but you could solve in this way putting a large number of characters that the reader is lost (it is similar en results section).

Author Response: We agree that the morphological description of the holotype is very detailed and dense. Most readers can simply skip over this section, because it is meant only for specialists. However, this detailed and dense description of the holotype is standard practice for descriptions of new species. Furthermore, the structure and format of this section closely follows other recent descriptions of new Abronia species, so that we fulfill the established conventions in this field of scientific study. The salient point is that more simple and readable coverage of the diagnostic characters that separate the new species from existing members of the genus is provided in the Diagnosis section of our manuscript. Hence, our manuscript already addresses the reviewer’s critique. 

(Reviewer #4): Methods-How many specimens were collected for this analysis? Do you have the number each specimens? (collection label number), if so, could you add in any part of the methods section.

Author Response: These details are provided in the first three paragraphs of the Results section. Because they are results, we provide them in the Results section instead of the Methods.

(Reviewer #4): Table 1-Format and size (bigger letters) of the letter should be better.

Author Response: The current formatting and fonts within the cells of the table meet the requirements of PLOS ONE. The contents of the table will be made more easily readable in the final formatting as part of the production process.

(Reviewer #4) Results, lines 484–490: I believe that more importance should be given to the area or regions in which the genus Abronia is distributed, which is more in the cloud forest and pine forest of the mountains, and to consider the deterioration of the environment, and therefore, the danger of its populations.

Author Response: In this paragraph, our goal is to discuss the specific distribution and ecology of the new species that we are proposing for recognition. This meets the core purpose of our paper, which is to provide information that relates to the new species. Deep coverage of the geographic distribution of, and conservation issues relating to, the entire genus Abronia is beyond the scope of our manuscript. However, in the second paragraph of our Introduction, we provide readers with a primer on these topics as they relate to Abronia more generally.

(Reviewer #4) Results, final paragraph: It is a long explanation about "conservation", but there is nothing concrete that you as authors propose the care and conservation of the landscape in which this new species lives and all the species that live in sympatria, need a greater vision on the conservation of the biodiversity.

Author Response: The purpose of this paragraph is to provide detailed, concrete justification for why we believe the proposed new species qualifies for listing as Threatened (Amenazada) by the Mexican federal government, in accordance with the specific criteria used by the government to make those decisions. We discuss and propose explicit conservation actions for the new species and its habitat in the concluding paragraph of our Discussion. 

(Reviewer #4): I would like at the end of discussion (conclusion) talk a little bit about conservation of the arboreal species, because the fragmentation of the forest (mountain cloud forest) by human is strong, many arboreal and terrestrial species from this environments, their populations are disappearing It is important to point out the care of the environment in these areas where the diversification of the species occurs, focusing on the conservation of biodiversity, specifically in the genus Abronia, whose species are arboreal; in these environments, the anthropic effect is ending the vegetation cover, eliminating the refuge(s) of the species.

Author Response: We have added several new citations to our concluding paragraph (and have modified our extensive discussion therein) to explicitly address the importance of protecting forests from land use change, in the context of the proposed new Abronia and the region of Chiapas where it lives.

(Reviewer #5, anonymous): In this study, the authors describe a new species of the arboreal lizard genus Abronia. Most species in this genus are endangered, including this new species. The text is well structured, and all the relevant characters for the description of the species were considered. Also, important information about the geographic distribution of the Abronia species from the state of Chiapas, Mexico, is provided. I believe this work deserves to be considered for its publication in PLoS ONE as it is.

Author Response: We are grateful for the reviewer’s favorable perspective regarding our manuscript.

(Reviewer #6, anonymous): This manuscript describes a new species of Arboreal Alligator lizard (Abronia) from the Northern highlands of Mexican Chiapas. The authors identify diagnostic combinations of scalation characters and geographic isolation as evidence for these animals belonging to a new species. The paper is well written with good figures, the description and diagnoses are well laid out and appropriate and, as Abronia species have famously restricted ranges, this almost certainly will prove to be a new species. However, if this is published as is, it leaves some shortcomings that will need to addressed up by future studies, and I worry that the authors have not done enough to warrant this being published in PlosOne.

Author Response: We appreciate these kind words and constructive critique of our manuscript. We hope that the addition of molecular evidence to our revised manuscript (discussed at length below) has sufficiently addressed the major shortcomings identified by the reviewer.

The authors employ De Quiroz’s evolutionary species concept and identify 1) fixed morphological differences between the new species and select congeners, and 2) geographic isolation as their delimitation methods. No molecular phylogenetics were attempted, despite tissue being available from the new species and the putative sister species (A.morenica), and mtDNA sequences are available on genbank from the geographically proximal A lythrochila, and the lissabronine A. frosti, as well as several other species (eg A. moreletii, A. graminea, A. monticola, A. cuchumatanus, A. gadovii, A. aurita, A. lythrochila, A. campbelli, A. fimbriata, A. sp., A. matudai, A. anzuetoi, A. mixteca, A. ornelasi, Gerrhonotus liocephalus, Barisia imbricata, A. chiszari, and A. oaxacae), not to mention the data from an unpublished Masters thesis from the anchor author, and a well sampled phylogenomic analysis that was also coauthored by the anchor author. I understand that that there are challenges to incorporating novel sequences into existing datasets, especially genomic ones, but if the authors were able to incorporate some genetic data it would really strengthen the manuscript and further untangle the Abronia relationships.

Author response: We thank the reviewer for pointing out the lack of molecular data in our original manuscript, which we agree was a shortcoming. We have now added to our description a molecular phylogeny for the genus Abronia, based on rigorous genomic ddRADseq analysis. This molecular phylogeny includes multiple samples of our proposed new species, its suspected sister species, and all other known Abronia species that occur nearby. This molecular phylogeny shows that the proposed new species of Abronia is a distinct lineage, although closely related to the allopatric Abronia morenica. Therefore, three independent lines of evidence (genomic, morphological, and biogeographical) indicate that our proposed new species of Abronia warrants formal recognition. 

The Authors assign the new species to a subgenus without any phylogenetic analysis to confirm this statement. Furthermore, the proposed assignment of the new species is based on a taxonomy that is has been shown, through molecular phylogenetics, to not consistently represent the evolutionary history of the group. The most recent phylogeny by Guitérrez-Rodríguez et al. 2020 suggests that many /most of the subgenera proposed by Campbell and Frost (1993) were paraphyletic, and even showed that Abronia as a whole was paraphyletic! Although the authors follow the taxonomic assessment of the 2020 paper and refer to the 11 clades, they still use the groupings of Campbell and Frost to describe the phylogenetic placement of the new species. They place it the subgenus Lissabronia, and point to its geographic isolation from all other members of the subgenus as evidence of its evolutionary isolation, dismissing the possibility that the new species could be associated with the geographically proximal A.lythrochila. Perhaps this new species is unambiguously attributable to Lissabronia (though the subgenus does lack autapomorphies-not synapomorphies as stated), but a phylogenetic assessment, even morphological, would help make that point.

Author Response: The reviewer makes excellent points, and we agree. With the addition of the molecular phylogeny to our paper, our tentative assignment of our proposed new species to the subgenus Lissabronia is more well supported because the genomic data resolve the proposed new species as sister to Abronia morenica. This finding is echoed in the morphological evidence. More conclusive resolution of the subgeneric assignment of the proposed new species will necessitate the collection of DNA samples from the other species that are currently assigned to Lissabronia (A. frosti, A. montecristoi, and A. salvadorensis), but such samples are not available to us and, to our knowledge, do not exist in collections. We also note that, consistent with morphology, the molecular phylogeny resolves the proposed new species as distantly related to A. lythrochila.

Furthermore, I expect that some ecological niche models would help confirm that the species is geographically isolated from the nearby A. lythrochila, especially using data layers that reveal vegetation cover, which seems to be an important factor in Abronia distribution.

Author Response: Echoing our previous answer, in our manuscript we now present both morphological and molecular evidence are concordant in showing that the proposed new species is not closely related to A. lythrochila. Whether or not the two species are, in fact, geographically isolated is thus somewhat of a moot point. Although we agree that ecological niche models could shed light on this point, such models require large presence or presence/absence datasets as inputs (generally, at least 30 locality points) to generate defensible outputs. Unfortunately, fewer than 20 locality points exist for A. lythrochila (see Figure 4 in our manuscript), and just 1 locality point exists for the proposed new species. We therefore consider ecological niche models to be premature for this particular biological system. 

In summary, I am sure that this species assignment will prove valid, but I am concerned that the reliance of scalation characters to delimit the species and assign to a subgenus is undermined by the inconsistencies the morphological and molecular phylogenies. The authors do have tissues for this and the putative sister species, and I would strongly recommend that they incorporate some kind of phylogenetic analysis into this description.

Author Response: We agree with the reviewer’s concern about our reliance on scalation characters in our original manuscript. As mentioned, we have now added a phylogenetic analysis to our paper that includes samples of both the proposed new species from Coapilla and its putative sister species, A. morenica. The results of this genomic dataset agree with the morphological and biogeographic evidence. Together, these three lines of evidence support the distinctiveness of the Abronia from Coapilla as a lineage that warrants recognition as a new species, albeit one closely related to A. morenica.

Author note to editor: In the second sentence of the third paragraph of our Introduction, we have added a citation to Hidalgo-García et al. (2023), because those authors present new data relevant to this sentence that was published after we submitted our manuscript. Additionally, in the second sentence of the fifth paragraph of the Distribution and Ecology section, we have revised the maximum known size of Abronia anzuetoi based on new findings published by Reyes et al. (2023), which we now cite in that sentence as well. We have updated the Literature Cited section of our revised manuscript accordingly. Thank you.

Comment from Editorial Team: Please ensure that your manuscript meets PLOS ONE's style requirements, including those for file naming.

Author Response: We have reviewed our manuscript and made the necessary edits to meet the style requirements of PLOS ONE. Thank you.

Comment from Editorial Team: We note that you have referenced (unpublished on page 25) which has currently not yet been accepted for publication. Please respond by return e-mail with a copy of your updated manuscript to include to remove this from your References and amend this to state in the body of your manuscript: (ie “Bewick et al. [Unpublished]”) as detailed online in our guide for authors

http://journals.plos.org/plosone/s/submission-guidelines#loc-reference-style. We can then upload this to your submission on your behalf.

Author Response: The unpublished reference has since been officially published, and hence in our revised manuscript we include the full citation for this now-published reference. Thank you.

Comment from Editorial Team: We note that Figure 4 in your submission contain [map/satellite] images which may be copyrighted. All PLOS content is published under the Creative Commons Attribution License (CC BY 4.0), which means that the manuscript, images, and Supporting Information files will be freely available online, and any third party is permitted to access, download, copy, distribute, and use these materials in any way, even commercially, with proper attribution. For these reasons, we cannot publish previously copyrighted maps or satellite images created using proprietary data, such as Google software (Google Maps, Street View, and Earth). For more information, see our copyright guidelines: http://journals.plos.org/plosone/s/licenses-and-copyright.

a. You may seek permission from the original copyright holder of Figure 4 to publish the content specifically under the CC BY 4.0 license. 

Maps at the CIA (public domain): https://www.cia.gov/library/publications/the-world-factbook/index.htmland
https://www.cia.gov/library/publications/cia-maps-publications/index.html

Author Response: Figure 4 (now Figure 8, in our revised manuscript) contains no copyrighted map/satellite images. All relevant layers were sourced from Natural Earth (public domain). We have added a sentence stating this in the caption of this Figure. If additional information is needed to resolve this point, please let us know. Thank you.

Comment from Editorial Team: Please include captions for your Supporting Information files at the end of your manuscript, and update any in-text citations to match accordingly. Please see our Supporting Information guidelines for more information: http://journals.plos.org/plosone/s/supporting-information. 

Author Response: We have now included captions for our Supporting Information at the end of our manuscript. Thank you.

---

## [Decision Letter · Decision Letter 1]

23 Oct 2023

PONE-D-22-34275R1Bridging the gap: A new species of arboreal Abronia (Squamata: Anguidae) from the Northern Highlands of Chiapas, MexicoPLOS ONE

Dear Dr. Clause,

Thank you for submitting your manuscript to PLOS ONE. After careful consideration, we feel that it has merit but does not fully meet PLOS ONE’s publication criteria as it currently stands. Therefore, we invite you to submit a revised version of the manuscript that addresses the points raised during the review process.

We look forward to receiving your revised manuscript.

Kind regards,

Tzen-Yuh Chiang

Academic Editor

PLOS ONE

Journal Requirements:

Reviewers' comments:

Reviewer's Responses to Questions

**Comments to the Author**

1. If the authors have adequately addressed your comments raised in a previous round of review and you feel that this manuscript is now acceptable for publication, you may indicate that here to bypass the “Comments to the Author” section, enter your conflict of interest statement in the “Confidential to Editor” section, and submit your "Accept" recommendation.

Reviewer #2: All comments have been addressed

Reviewer #3: All comments have been addressed

Reviewer #5: All comments have been addressed

2. Is the manuscript technically sound, and do the data support the conclusions?

Reviewer #2: Yes

Reviewer #3: Partly

Reviewer #5: Yes

3. Has the statistical analysis been performed appropriately and rigorously? 

Reviewer #2: Yes

Reviewer #3: Yes

Reviewer #5: Yes

4. Have the authors made all data underlying the findings in their manuscript fully available?

Reviewer #2: Yes

Reviewer #3: No

Reviewer #5: Yes

5. Is the manuscript presented in an intelligible fashion and written in standard English?

Reviewer #2: Yes

Reviewer #3: Yes

Reviewer #5: Yes

6. Review Comments to the Author

Reviewer #2: You have done a good job thoughtfully responding to my previous suggestions. This is a nice contribution to our understanding of the herpetofauna of Mexico, and in particular the relatively hidden species in this genus.

Reviewer #3: I am glad to see that authors have added genomic analyses to support their conclusion. I encourage them to make their data freely available. ddRAD is bit of overkill to such questions, more then enough, but it has reproducibility issues and sometimes old good mtDNA locus would be of a great value for future species identification (not everybody has access to ddRAD) - could you add it too for at least holotype?

Last not least, conStruct is a new software for me, but if it is clustering similar to Structure and other programs, running it with 6 specimens and grouping them into 3 species, one of them is represented by only one specimen seems not correct, moreover resulting plot also shows rather two well supported groups there, not three.

Reviewer #5: The authors have adequately addressed all the comments made by the reviewers. I therefore believe that this manuscript is ready for its publication in PLoS ONE

7. PLOS authors have the option to publish the peer review history of their article (what does this mean?). If published, this will include your full peer review and any attached files.

Reviewer #2: No

Reviewer #3: No

Reviewer #5: No

---

## [Author Response · Author response to Decision Letter 1]

6 Nov 2023

Journal Requirements: Please review your reference list to ensure that it is complete and correct. If you have cited papers that have been retracted, please include the rationale for doing so in the manuscript text, or remove these references and replace them with relevant current references. Any changes to the reference list should be mentioned in the rebuttal letter that accompanies your revised manuscript. If you need to cite a retracted article, indicate the article’s retracted status in the References list and also include a citation and full reference for the retraction notice.

Author Response: We have double-checked our reference list, and we affirm that it is complete and correct. We cite no papers that have been retracted. We have also updated an early-view paper (reference #37, Scarpetta and Ledesma) that now has a volume, issue, and pagination information.

Reviewer #2: You have done a good job thoughtfully responding to my previous suggestions. This is a nice contribution to our understanding of the herpetofauna of Mexico, and in particular the relatively hidden species in this genus.

Author Response: We appreciate the reviewer’s favorable assessment of our revised manuscript.

Reviewer #3: I am glad to see that authors have added genomic analyses to support their conclusion. I encourage them to make their data freely available.

Author Response: We agree with the reviewer’s excellent request to make our genomic data freely available. In our Appendix 2, we now include GenBank numbers for all genomic sequence data included in our phylogenetic analyses. We have also cited the BioProject accession number for our GenBank upload of the new sequence data we produced as part of our study (lines 155–157).

Reviewer #3: ddRAD is bit of overkill to such questions, more then enough, but it has reproducibility issues and sometimes old good mtDNA locus would be of a great value for future species identification (not everybody has access to ddRAD) - could you add it too for at least holotype?

Author Response: We thank the reviewer for this comment and suggestion. Currently, in Genbank there are recently generated ddRADseq data for most Abronia species (Gutiérrez-Rodríguez et al. 2021; García-Vázquez et al. 2022; Gutiérrez-Rodríguez et al. 2022) and anyone can access them. Given that the most complete phylogeny for this group was obtained with ddRADseq data (Gutiérrez-Rodríguez et al. 2021), we prefer to continue obtaining genomic data for the missing species with the aim of building a more complete and robust phylogeny. It is true that assemblies of aligned sequences generated from ddRADseq data (e.g., for phylogenetic analyses) may vary depending on several parameters. That is precisely why we described both the laboratory protocol we followed to generate the data, and the pipeline and parameter settings that we used to generate sequence alignments. This guarantees that if the same pipeline and parameter settings are used, the same sequence alignment will be obtained. We did the same for all the analyses we performed with those data. Regarding the reviewer’s suggestion to include data from a mtDNA locus, we appreciate this idea for further enriching our research. However, because the proposed new species is easily identifiable morphologically from all congeneric species (as shown in Figure 8, Table 1, and in our Comparisons section on lines 421–460), it is not cryptic and hence mtDNA sequence data is unnecessary to identify it. Specimens of the new species can be confidently identified from simple examination of external morphology, which is accessible to anyone. We respectfully consider mtDNA data unnecessary to rigorously delimit or identify species when (as is the case under consideration here) phylogenomic, morphological, and biogeographic data are concordant in supporting species recognition and when phylogenomic data also allow inference of robust phylogenetic hypotheses.

Reviewer #3: Last not least, conStruct is a new software for me, but if it is clustering similar to Structure and other programs, running it with 6 specimens and grouping them into 3 species, one of them is represented by only one specimen seems not correct, moreover resulting plot also shows rather two well supported groups there, not three.

Author Response: We are aware that clustering programs are mainly used for population genetics where a greater number of samples are required. However, they have also been used for species delimitation. Our conStruct analysis provides another source of evidence and, despite the sampling limitations, provides data (see below) that supports and complements the other sources of evidence that we present. Regarding the small number of specimens, only one genetic sample of Abronia ornelasi is currently available. Because A. ornelasi has not been found by scientists since the 1980s, despite exhaustive searching at the only known locality by the authors and others, obtaining additional samples is not easily achievable. Having just one A. ornelasi sample included in our conStruct analysis is therefore unavoidable at this time. Importantly, as explained in lines 297–308 of our manuscript, and as shown in Figure 3, A. ornelasi and the proposed new species draw their ancestry from different layers. This clearly indicates that they are not conspecific. Furthermore, as also explained in lines 297–308, the conStruct analysis indicates that the best model is K = 3, not K = 2 as suggested by the reviewer. We agree with the reviewer that, as shown in Figure 3, the proposed new species and A. morenica draw most of their ancestry from the same layer. However, the samples of the putative new species also draw about one-third of their ancestry from a different layer. We thus stand by the results of the conStruct analysis and our interpretation of those results as being consistent with the recognition of the newly proposed species of Abronia. We also transparently acknowledge, on lines 303–308, that the conStruct results suggest that the proposed new species diverged relatively recently from its sister species A. morenica following the geographic separation of these lineages.

Reviewer #5: The authors have adequately addressed all the comments made by the reviewers. I therefore believe that this manuscript is ready for its publication in PLoS ONE.

Author Response: We are grateful for the reviewer’s judgement that our revised manuscript is ready for publication. 

Author Note to Editor: In addition to the abovementioned changes and responses, we also wish to acknowledge a few minor alterations and additions that we made to the text of our manuscript. These changes, which do not affect our analyses, results, or conclusions, are as follows: (1) changing the phrase “putative new species” to “presumed new species” throughout our manuscript, so that we use a more widely-recognized and less jargony phrase; (2) presenting an alternative, citation-supported spelling and translation of the name Cuñemo in our Etymology section (lines 654–664). Lastly, we have ensured that all changes made to the manuscript are reflected in the updated Spanish-language translation of our entire manuscript (Appendix S1), for continuity between those texts.

---

## [Decision Letter · Decision Letter 2]

20 Nov 2023

Bridging the gap: A new species of arboreal Abronia (Squamata: Anguidae) from the Northern Highlands of Chiapas, Mexico

PONE-D-22-34275R2

Dear Dr. Clause,

We’re pleased to inform you that your manuscript has been judged scientifically suitable for publication and will be formally accepted for publication once it meets all outstanding technical requirements.

Kind regards,

Tzen-Yuh Chiang

Academic Editor

PLOS ONE

Additional Editor Comments (optional):

Reviewers' comments:

Reviewer's Responses to Questions

**Comments to the Author**

1. If the authors have adequately addressed your comments raised in a previous round of review and you feel that this manuscript is now acceptable for publication, you may indicate that here to bypass the “Comments to the Author” section, enter your conflict of interest statement in the “Confidential to Editor” section, and submit your "Accept" recommendation.

Reviewer #2: All comments have been addressed

Reviewer #3: All comments have been addressed

2. Is the manuscript technically sound, and do the data support the conclusions?

Reviewer #2: Yes

Reviewer #3: Yes

3. Has the statistical analysis been performed appropriately and rigorously? 

Reviewer #2: Yes

Reviewer #3: Yes

4. Have the authors made all data underlying the findings in their manuscript fully available?

Reviewer #2: Yes

Reviewer #3: Yes

5. Is the manuscript presented in an intelligible fashion and written in standard English?

Reviewer #2: Yes

Reviewer #3: Yes

6. Review Comments to the Author

Reviewer #2: The authors had already addressed my previous comments to my satisfaction in the previous revision. I have examined their responses to the other reviewers' comments on this version and found their responses compelling and satisfactory.

Reviewer #3: (No Response)

7. PLOS authors have the option to publish the peer review history of their article (what does this mean?). If published, this will include your full peer review and any attached files.

Reviewer #2: No

Reviewer #3: **Yes: **Oleksandr Zinenko

---

## [Editor Report · Acceptance letter]

7 Dec 2023

PONE-D-22-34275R2 

Bridging the gap: A new species of arboreal *Abronia* (Squamata: Anguidae) from the Northern Highlands of Chiapas, Mexico 

Dear Dr. Clause:

I'm pleased to inform you that your manuscript has been deemed suitable for publication in PLOS ONE. Congratulations! Your manuscript is now with our production department. 

Kind regards, 

on behalf of

Dr. Tzen-Yuh Chiang 

Academic Editor

PLOS ONE